# The role of the NMD factor UPF3B in olfactory sensory neurons

Kun Tan[1†], Samantha H Jones[1†], Blue B Lake[2], Jennifer N Chousal[1], Eleen Y Shum[1], Lingjuan Zhang[3], Song Chen[2], Abhishek Sohni[1], Shivam Pandya[1], Richard L Gallo[3], Kun Zhang[2], Heidi Cook-Andersen[1,4], Miles F Wilkinson[1,5]*

[1]Department of Obstetrics, Gynecology, and Reproductive Sciences, School of Medicine University of California, San Diego, San Diego, United States; [2]Department of Bioengineering, University of California, San Diego, San Diego, United States; [3]Department of Dermatology, University of California, San Diego, San Diego, United States; [4]Division of Biological Sciences, University of California, San Diego, San Diego, United States; [5]Institute of Genomic Medicine, University of California, San Diego, San Diego, United States

*For correspondence:
mfwilkinson@health.ucsd.edu

[†]These authors contributed equally to this work

Competing interests: The authors declare that no competing interests exist.

**Abstract** The UPF3B-dependent branch of the nonsense-mediated RNA decay (NMD) pathway is critical for human cognition. Here, we examined the role of UPF3B in the olfactory system. Single-cell RNA-sequencing (scRNA-seq) analysis demonstrated considerable heterogeneity of olfactory sensory neuron (OSN) cell populations in wild-type (WT) mice, and revealed that UPF3B loss influences specific subsets of these cell populations. UPF3B also regulates the expression of a large cadre of antimicrobial genes in OSNs, and promotes the selection of specific olfactory receptor (*Olfr*) genes for expression in mature OSNs (mOSNs). RNA-seq and Ribotag analyses identified classes of mRNAs expressed and translated at different levels in WT and *Upf3b*-null mOSNs. Integrating multiple computational approaches, UPF3B-dependent NMD target transcripts that are candidates to mediate the functions of NMD in mOSNs were identified in vivo. Together, our data provides a valuable resource for the olfactory field and insights into the roles of NMD in vivo.

## Introduction

Nonsense-mediated RNA decay (NMD) is a conserved pathway originally discovered by virtue of its ability to degrade aberrant RNAs harboring premature termination codons (PTCs) and thus protect cells from truncated, potentially toxic, dominant-negative proteins (*Chang et al., 2007*; *Conti and Izaurralde, 2005*; *Kurosaki et al., 2019*; *Lykke-Andersen and Jensen, 2015*; *Palacios, 2013*). Subsequently, it was discovered that NMD degrades subsets of normal RNAs, with loss or disruption of NMD leading to dysregulation of 5–20% of the normal transcriptome in species spanning the phylogenetic scale (*Chan et al., 2007*; *He et al., 2003*; *Mendell et al., 2004*). This discovery raised the possibility that the function of NMD extends beyond quality control. This notion has been supported by scores of subsequent studies showing that NMD factors are critical for many fundamental processes, including development, differentiation, cell proliferation, the integrated stress response, the unfolded protein response, and autophagy (*Chang et al., 2007*; *Karam et al., 2015*; *Kurosaki et al., 2019*; *Nasif et al., 2018*).

NMD is well-studied at the biochemical level, with over 15 proteins known to be involved in this pathway (*Chang et al., 2007*; *Kurosaki et al., 2019*). Three of these proteins—UPF1, UPF2, and UPF3—are present in all eukaryotes and considered to be the core NMD factors (*Conti and Izaurralde, 2005*). UPF1 is an RNA helicase that forms a complex with the adaptor proteins UPF2 and UPF3. In vertebrates, UPF3 is encoded by two paralogs: *UPF3A* (also called '*UPF3*') and *UPF3B* (also

called 'UPF3X'). UPF3A serves as a weak NMD factor and NMD repressor, while UPF3B is a NMD branch-specific factor that stimulates NMD (*Chan et al., 2007*; *Shum et al., 2016*). UPF3B directly binds to the exon-junction complex (EJC), a large multi-subunit complex recruited to RNAs just upstream of exon-exon junctions after RNA splicing (*Woodward et al., 2017*). The EJC triggers NMD when allowed to interact with other NMD factors. Evidence suggests that EJCs are displaced by the ribosome during the pioneer round of translation, and thus only EJCs deposited downstream of the stop codon defining the main open-reading frame (ORF) are able to elicit NMD (*Dostie and Dreyfuss, 2002*). Ribosomes would also be predicted to displace the last EJC when the termination codon resides ~50 nucleotides or less upstream of the last exon-exon junction, based on the length of the EJC and ribosome footprints. This has led to the '−50-nt boundary rule,' an empirically veri-fied dictum which states that only in-frame stop codons further than ~50 nt upstream of the last exon-exon junction elicit NMD (*Nagy and Maquat, 1998*). While there are exceptions to this −50-nt boundary rule (*Carter et al., 1996*), it reliably predicts a large proportion of EJC-dependent NMD target mRNAs (*Boehm et al., 2014*; *Gehring et al., 2005*; *Hurt et al., 2013*). NMD can be triggered by other molecular signals in addition to downstream EJCs. For example, long 3'-untranslated regions (UTRs) and short ORFs upstream of the main ORF (uORFs) can, in some cases, trigger NMD in an EJC-independent manner (*Barrett et al., 2012*; *Bühler et al., 2006*; *Chang et al., 2007*; *Hurt et al., 2013*; *Kebaara and Atkin, 2009*; *Rebbapragada and Lykke-Andersen, 2009*).

While considerable progress has been made in understanding the molecular features that elicit NMD, we are still largely in the dark with regard to which transcripts are targeted for rapid decay. It is critical to define such NMD target mRNAs in order to begin to unravel the molecular mechanisms by which NMD influences biological processes. A particularly large gap in the field is the identity of NMD targets in vivo.

Many lines of evidence suggests that NMD is not a single linear pathway but instead consists of several branches, each of which depends on different factors and promotes the decay of different sets of transcripts (*Chan et al., 2007*; *Gehring et al., 2005*; *Mabin et al., 2018*). In this report, we focus on the UPF3B-dependent branch of NMD, which has been shown to be important for the ner-vous system. Pedigree analysis of numerous families harboring mutations in the *UPF3B* gene have demonstrated that both nonsense and missense mutations cause intellectual disability in humans (*Nguyen et al., 2014*; *Tarpey et al., 2007*). Humans with *UPF3B* mutations also commonly have autism, schizophrenia, and/or attention-deficit/hyperactivity disorder (*Nguyen et al., 2014*; *Tarpey et al., 2007*). To understand the underlying mechanism for these behavioral defects, we gen-erated *Upf3b*-deficient mice (*Huang et al., 2011*; *Karam et al., 2015*). These *Upf3b*-null mice suffer from specific learning and memory deficits, including fear-conditioned learning, and thus replicate some aspects of the behavioral defects in UPF3B-deficient humans (*Huang et al., 2018*). In part, these behavioral defects may stem from abnormal neural connectivity, as cortical pyramidal neurons from *Upf3b*-null mice undergo impaired dendritic spine maturation in vivo (*Huang et al., 2018*). Fur-thermore, cultured UPF3B-depleted neural cells have subtle dendrite outgrowth defects (*Jolly et al., 2013*), and expression of UPF3B mutants reduces neurite branching (*Alrahbeni et al., 2015*). The behavioral defects in *Upf3b*-null mice may also result from neural differentiation and/or maturation defects that were uncovered using loss-of-function approaches in neural precursor cells in vitro, or by forced expression of UPF3B mutants in cell lines in vitro (*Alrahbeni et al., 2015*; *Huang et al., 2018*; *Jolly et al., 2013*).

In this communication, we examine the role of UPF3B in the olfactory system, a useful model for studying neural development and function. There is also considerable clinical interest in the olfactory system, as olfactory defects predict the later onset of numerous CNS disorders, including Parkinson's and Alzheimer's disease (*Doty, 2012*). Olfactory dysfunction also strongly associates with autism (*Rozenkrantz et al., 2015*). The olfactory epithelium (OE) retains a life-long capacity for neurogene-sis and harbors a robust regeneration system that responds to injury (*Whitman and Greer, 2009*). Importantly, the olfactory system is much simpler than the CNS. Mature olfactory sensory neurons (mOSNs) develop via a relatively simple linear pathway involving horizontal basal cells (HBCs), glo-bose basal cells (GBCs), and immature olfactory sensory neurons (iOSNs). Both HBCs and GBCs are stem cells, but the two types have different roles (*Schwob et al., 2017*). HBCs are reserve stem cells, as they are normally quiescent and only undergo proliferative expansion in response to OE injury (*Peterson et al., 2019*). In contrast, GBCs are a heterogeneous cell population that consists of con-stitutively active stem cells as well as progenitors (*Schwob et al., 2017*). Lineage-tracing analysis

and single-cell RNA sequencing (scRNA-seq) analysis have shown that after proliferative expansion, HBCs and GBCs give rise to iOSNs, which are responsible for undergoing maturation (*Fletcher et al., 2017*). Of note, iOSNs share markers with another OSN stage called 'immediate neural precursors (INPs)." Given the ambiguity of the nomenclature, we will refer to cells with either INP or iOSN characteristics as iOSNs. iOSNs ultimately differentiate into mOSNs, which send an axon to neurons in the glomeruli region of the olfactory bulb, relaying olfactory information from the outside world to the CNS. mOSNs recognize odorants through chemosensory receptors, including olfactory receptors (OLFRs), members of the G-protein-coupled receptor super-family, as well as trace amine–associated receptors, guanylate cyclases, and members of the membrane-spanning 4-pass A gene family (*Bear et al., 2016*; *Saraiva et al., 2019*).

To gain insight into the nature of the cells in the OE and their developmental relationships, recent studies have performed transcriptome profiling using whole OE, pools of sorted OSNs, single OSNs, or single OE cells (*Fletcher et al., 2017*; *Ibarra-Soria et al., 2014*; *Saraiva et al., 2015*; *Saraiva et al., 2019*; *Tan et al., 2015*). These studies have revealed new OE cell subsets, inferred the developmental pathways of both OSN and non-neural OE cells, defined classes of genes exhibiting enriched expression and unique patterns of expression in different OE subsets, and revealed the expression patterns and dynamics of OLFRs during OSN development and in individual mOSNs. These studies have also advanced our understanding of mammalian olfaction evolution.

In this study, we ascertain whether the NMD factor, UPF3B, has roles in the olfactory system. Using scRNA-seq and RNA-seq analyses, we obtained evidence that UPF3B influences the frequency of specific OSN subsets, broadly suppresses the expression of immune genes in OSNs, and shapes the *Olfr* gene repertoire. We also identified high-confidence NMD target mRNAs in vivo that are candidates to act downstream of UPF3B in mOSNs. As part of our analysis, we also provide new cellular and molecular information on WT OSNs and their development in vivo. Our findings in *Upf3b*-null NMD-deficient mice introduce a useful biological system to understand the role of RNA metabolism in neurons, and our scRNA-seq, RNA-seq, and RiboTag datasets are new resources that can be used by the olfactory field.

## Results

### UPF3B-regulated genes in mOSNs

To assess whether NMD-deficient *Upf3b*-null mice have an olfactory defect, we measured their weight during their growth phase. This follows from the fact that newborn mice are blind and therefore depend on the olfactory system to initiate milk suckling for survival (*Logan et al., 2012*). We quantified the weight of *Upf3b*-null and littermate WT mice and found that *Upf3b*-null mice have a statistically significant postnatal weight deficit (p<0.05; *Figure 1A*). The weight deficit occurs soon after birth, becomes progressively worse during postnatal development, and is corrected after reaching adulthood. This specific pattern of weight loss is characteristic of mice harboring a partial olfactory defect (*Riera et al., 2017*). In contrast, newborn mice that completely lack sense of smell are incapable of sensing their source of milk and die soon after birth (*Hongo et al., 2000*). As further evidence that *Upf3b*-null mice have an olfactory defect, we found that HBC, iOSN and mOSN marker genes (*Krt5, Gap43*, and *Gnal*, respectively) exhibited significantly decreased expression in *Upf3b*-null as compared to WT OE (*Figure 1B*). We followed up by testing *Upf3b*-null mice for evidence for specific olfactory deficits and observed trends but did not observe statistically significant effects (*Figure 1—figure supplement 1A*), providing further evidence of a partial olfactory defect.

Given that mOSNs are the functional units of the OE, we next focused our attention on these cells. We identified UPF3B-regulated genes in mOSNs by performing RNA-seq analysis on FACS-purified mOSNs (YFP+ cells) from R26-eYFP; *Omp*-Cre mice (*Figure 1—figure supplement 1B*). Four samples were analyzed from each genotype (*Figure 1—figure supplement 2A* and *Supplementary file 1*). The expression of OSN precursor/OSN canonical markers are shown in *Figure 1—figure supplement 2D*. RNA-seq analysis identified 235 differentially expressed genes between *Upf3b*-null and WT mOSNs (q < 0.05) (*Figure 1C* and *Supplementary file 2*). We validated our RNA-seq analysis by qPCR analysis and immunofluorescence (*Figure 1—figure supplement 2B, C*).

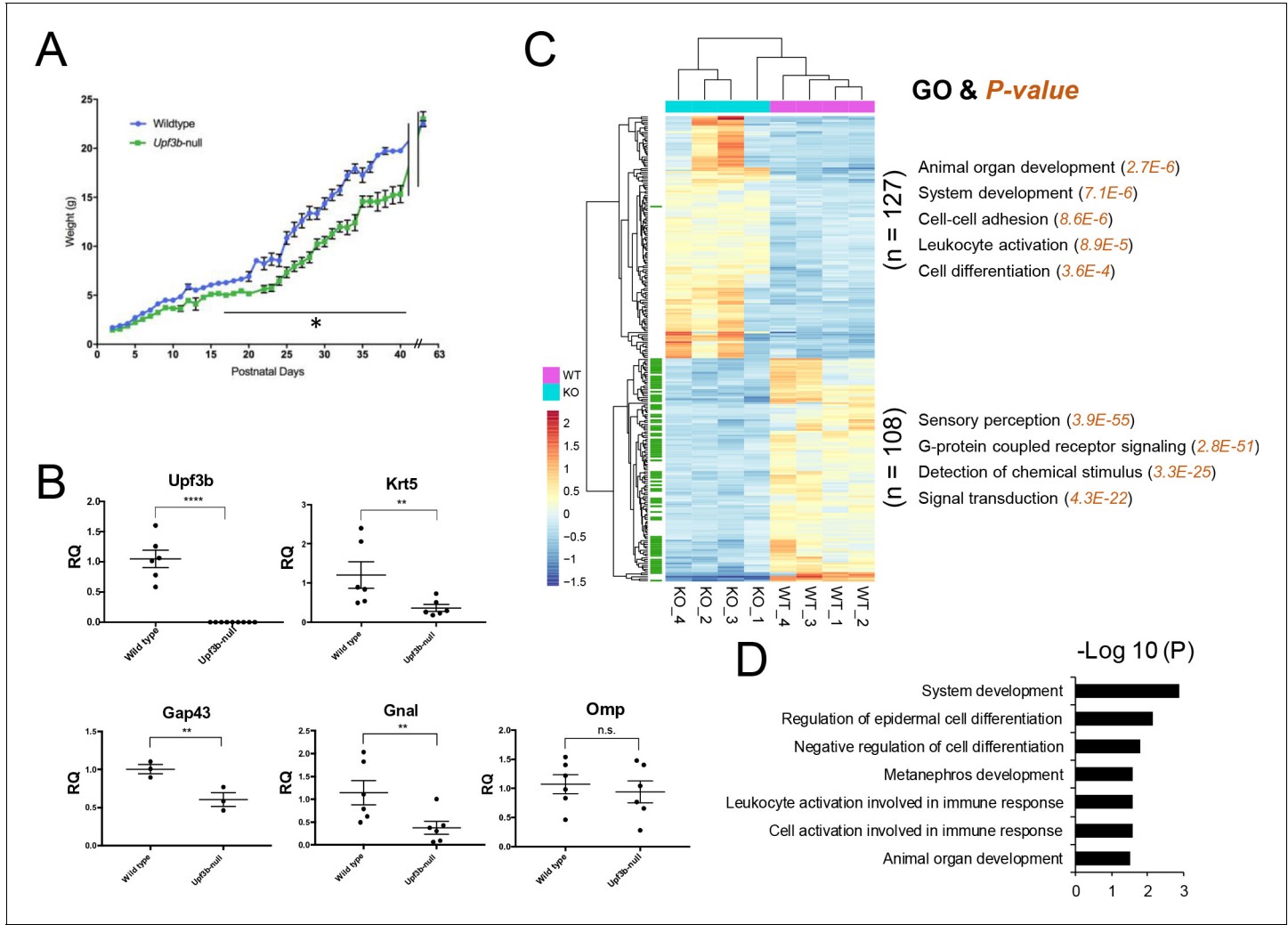

**Figure 1.** Identification of UPF3B-regulated genes and NMD target genes in the olfactory system. (**A**) The weight of *Upf3b*-null vs. WT (wild type) mice at the indicated time points. *Upf3b*-null mice gain weight slowly during postnatal development but then reach the weight of WT mice at the last time point (9 weeks), a pattern indicative of a partial olfactory defect. *, p<0.05. (**B**) qPCR analysis of olfactory marker genes in *Upf3b*-null and WT OE (n = 6). **, p<0.01; ****, p<0.0001. (**C**) Heatmap of genes differentially expressed in mOSNs from *Upf3b*-null (KO) vs. WT mice (four biological replicates from each are shown). Row names labeled as green are *Olfr* genes. Right, the most statistically significant GO terms associated with upregulated genes (top) and downregulated genes (bottom) after *Upf3b* loss. (**D**) A list of most statistically enriched GO terms associated with the 52 high-confidence UPF3B-dependent NMD target mRNAs we identified in mOSNs.

The online version of this article includes the following figure supplement(s) for figure 1:

**Figure supplement 1.** Upf3b-null mice behavior and purified mOSNs.

**Figure supplement 2.** UPF3B-regulated genes in mOSNs.

Among the 127 upregulated genes were several involved in neurogenesis, including *Lrp2*, *Hk2*, *Notch2*, *Gdf11*, *Fos*, *Ptch1*, and *Spry2*. Gene ontology (GO) analysis revealed enrichment for 'organ/system development,' 'cell-cell adhesion,' 'leukocyte activation,' and 'cell differentiation/proliferation' functions (*Figure 1C*). In contrast, the 108 downregulated genes were most enriched for GO functions associated with olfaction: 'sensory perception,' 'G-proteins,' 'detection of chemical stimulus,' and 'signal transduction.' Indeed, we found that the majority (78 out of 108) of these significantly downregulated genes are *Olfr* genes (marked in green in *Figure 1C*; the expression of all *Olfr* genes in *Upf3b*-null and control mOSNs is shown in *Figure 1—figure supplement 2E* and *Supplementary file 1*). We follow-up on this surprising finding below. Other genes downregulated in *Upf3b*-null mOSNs include those involved in CNS synaptic transmission (*Slc17a6*), chromatin remodeling (*Chd1*), and sensory neuronal plasticity (*Cwc22*) (*Supplementary file 2*).

## Identification of NMD target mRNAs in mOSNs

NMD is thought to influence biological systems by virtue of its ability to promote the decay of specific subsets of mRNAs (*Lykke-Andersen and Jensen, 2015*). As described in the introduction, there is dearth of knowledge regarding the identity of such NMD target RNAs, particularly in cells in their normal in vivo context. Our RNA-seq analysis of purified mOSNs from *Upf3b*-null and WT mice provided an opportunity to identify in vivo direct NMD targets. Because NMD is a negative regulatory pathway (it degrades its targets), the 127 RNAs *upregulated* in *Upf3b*-null mOSNs are candidates to be direct NMD targets. Among them, we found that 73 had at least one of the well-established molecular features known to elicit NMD, including an exon-exon junction >50 nt downstream of the main ORF (dEJ) (*Table 1*; see the Introduction for an explanation of NMD-inducing features [NIFs]). Thus, these 73 mRNAs are strong candidates to be UPF3B-dependent NMD target mRNAs in mOSNs.

Given that NMD degrades its target RNAs, this predicts that its targets should be stabilized after inactivation of UPF3B. Thus, we measured the stability of the 127 mRNAs upregulated in *Upf3b*-null mOSNs using a method that infers RNA stability based on pre-mRNA and steady-state mRNA levels (*Alkallas et al., 2017*). This method revealed that 82 of 127 upregulated genes encode mRNAs stabilized in *Upf3b*-null mOSNs as compared to WT mOSNs (*Supplementary file 2*). Of these 82 stabilized and upregulated mRNAs, 52 have at least 1 of the 3 well-established NIFs (*Table 1*), and thus we classified these 52 mRNAs as high-confidence mOSN NMD targets. The statistically enriched GO biological functions of the proteins encoded by these 52 mRNAs are listed in *Figure 1D*.

**Table 1.** UPF3B-dependent NMD target mRNAs in mOSNs.

| Symbol | log2FC (KO/WT) | Padj | dEJ | uORF | 3'UTR length | Symbol | log2FC (KO/WT) | Padj | dEJ | uORF | 3'UTR length |
|---|---|---|---|---|---|---|---|---|---|---|---|
| Prelid3a | 1.099967 | 0.003745 | YES | NO | 1572 | Fmo2 | 2.02039 | 0.014815 | NO | NO | 2411 |
| 1700025G04Rik | 0.662926 | 0.012989 | NO | YES | 8870 | Gab2 | 0.98414 | 0.003018 | NO | NO | 3927 |
| 6030419C18Rik | 0.73232 | 0.036112 | NO | YES | 55 | Gdf11 | 1.429234 | 0.005353 | NO | NO | 2811 |
| 9330159F19Rik | 0.542375 | 0.017617 | NO | YES | 3408 | Gldn | 2.115908 | 0.045841 | NO | NO | 2970 |
| Adcy6 | 2.587005 | 0.002078 | NO | YES | 2356 | Hk2 | 2.296045 | 0.033161 | NO | NO | 2285 |
| Cdh24 | 1.560901 | 0.001303 | NO | YES | 121 | Lbh | 1.417311 | 0.024315 | NO | NO | 2498 |
| Fam84b | 0.719841 | 0.001704 | NO | YES | 3969 | Luc7l | 0.492061 | 8.19E-05 | NO | NO | 3738 |
| Inpp5f | 1.178064 | 0.043839 | NO | YES | 949 | Map3k9 | 0.79841 | 0.021555 | NO | NO | 1029 |
| Lrp2 | 2.504276 | 0.008534 | NO | YES | 1305 | Msrb3 | 1.668851 | 0.033388 | NO | NO | 2972 |
| Mafg | 0.577538 | 0.046713 | NO | YES | 4167 | Neurl3 | 1.966306 | 0.00546 | NO | NO | 1763 |
| Plxnc1 | 2.322167 | 0.048567 | NO | YES | 2320 | Notch2 | 1.68375 | 0.047733 | NO | NO | 2917 |
| Prdm4 | 0.420203 | 0.027945 | NO | YES | 1160 | Plekha5 | 0.608634 | 0.004216 | NO | NO | 3461 |
| Ptch1 | 0.768864 | 0.01088 | NO | YES | 3205 | Rab43 | 1.033148 | 0.0151 | NO | NO | 3737 |
| Ptger2 | 3.032221 | 0.035664 | NO | YES | 1825 | Rac2 | 3.029035 | 0.038392 | NO | NO | 2319 |
| Sash3 | 2.352656 | 0.033245 | NO | YES | 1309 | Raver2 | 1.921185 | 0.027779 | NO | NO | 1892 |
| Serpinb11 | 1.991555 | 0.002719 | NO | YES | 468 | Rflnb | 0.755198 | 0.017617 | NO | NO | 2716 |
| Snx33 | 1.512032 | 0.012417 | NO | YES | 1258 | Sik1 | 2.027635 | 1.48E-06 | NO | NO | 2035 |
| Zfp36 | 1.802697 | 0.025165 | NO | YES | 774 | Slc38a6 | 1.21151 | 0.025847 | NO | NO | 1512 |
| Agap2 | 1.264604 | 0.00099 | NO | NO | 1357 | Slc5a1 | 2.563582 | 0.00527 | NO | NO | 1868 |
| Aox2 | 1.36834 | 0.018035 | NO | NO | 1640 | Swap70 | 1.863436 | 0.009993 | NO | NO | 2169 |
| Atp10d | 3.315656 | 0.017617 | NO | NO | 2384 | Tgm2 | 2.395934 | 0.042993 | NO | NO | 1399 |
| Bhlhe40 | 1.435423 | 0.000192 | NO | NO | 1593 | Themis2 | 3.496025 | 0.015464 | NO | NO | 1053 |
| Btg2 | 1.281148 | 0.000173 | NO | NO | 2199 | Tmprss2 | 2.167673 | 0.005867 | NO | NO | 1456 |
| Cybrd1 | 2.372842 | 0.002733 | NO | NO | 4269 | Tob2 | 0.667082 | 0.001453 | NO | NO | 2459 |
| Cyth4 | 2.162221 | 0.045105 | NO | NO | 1455 | Ywhag | 0.644673 | 0.017707 | NO | NO | 2586 |
| Ermn | 1.686519 | 0.005793 | NO | NO | 2641 | Zcchc6 | 0.512078 | 0.003018 | NO | NO | 1346 |

To determine whether these high-confidence NMD target mRNAs correspond to known NMD targets, we assembled a list of likely mouse NMD substrates defined by previous studies (*Supplementary file 3*). To qualify to be in this list, the RNA must have at least one known NMD-inducing feature (NIF) (*Palacios, 2013*) and experimental evidence from at least one assay that it is an NMD substrate (e.g. high UPF1 occupancy or upregulation and/or stabilization in response to NMD-factor depletion). We found that 11 of these previously defined likely mouse NMD target mRNAs overlapped with the 52 high-confidence targets identified in our study: *Atp10d, Lbh, Slc38a6, Tgm2, Notch2, Ywhag, Luc7l, Ptch1, 1700025G04Rik, Ptger2,* and *Msrb3*. Of note, it is not surprising that only a proportion of the upregulated mRNAs we identified in NMD-deficient mOSNs are previously known NMD targets, as NMD target mRNAs can be tissue-, cell type-, and NMD factor-specific (*Huang et al., 2011*). The list of previously defined candidate NMD targets that we compared with were defined in non-neuronal tissues and cell lines made deficient in NMD by knocking down or eliminating factors other than UPF3B (*Supplementary file 3*).

## The mOSN transcriptome and translome

We next determined the translation rate of mRNAs in mOSNs, both as a resource for the field and to address the relationship of NMD with translation in vivo. We assayed the translation rate of mRNAs in mOSNs using RiboTag mice, which express an epitope-tagged ribosomal protein, RPL22$^{HA}$, which is incorporated into actively translating ribosomes specifically in cells expressing CRE (*Sanz et al., 2009*). Immunoprecipitation (IP) of the cell lysates of interest with an HA antibody purifies the ribosome-associated mRNAs (*Figure 2A*, left) with an efficiency associated with polysome density (*Hornstein et al., 2016*). To examine ribosome density specifically in WT mOSNs, we isolated RiboTag-labeled mRNA from the OE of *RiboTag; Omp*-Cre mice and performed RNA-seq analysis. As a validation of cell-type specificity, we found that IP of OE lysates with the HA antisera enriched for the mOSN marker, *Omp*, whereas these lysates were depleted of the HBC and GBC markers, *Krt5* and *Lgr5*, respectively (*Figure 2A*, right). We then elucidated inferred translation efficiency (TE) for all expressed mRNAs in mOSNs – the 'mOSN translome' – by calculating the ratio of the IP signal from the RiboTag mice OE lysates over mOSN steady-state mRNA level, the latter determined as described above (*Supplementary file 2*).

Given that 3'UTR length has been shown to influence translation rates in cultured cells (*Spies et al., 2013*), we examined the relationship of 3'UTR length and TE in mOSNs in vivo. We found that mOSN mRNAs harboring 3'UTRs of >2 kb have much higher average TE than mOSN mRNAs harboring shorter 3'UTRs (*Figure 2B*). Highly translated mOSN mRNAs have an average 3'UTR length of ~1.8 kb, while lowly translated mOSN mRNAs have an average 3'UTR length of only ~0.9 kb (*Figure 2C*).

To assess the potential functional relevance of translation, we binned WT mOSN mRNAs into three groups: high (top 30%), medium (middle 40%), and low (bottom 30%) (*Supplementary file 2*). We also binned WT mOSN mRNAs into three groups based on their steady-state level (*Supplementary file 2*), allowing us to place mOSN mRNAs into the nine categories shown in *Figure 2D*. GO analysis revealed that category #1—which is mRNAs expressed at high level that are also highly translated—encode proteins that tend to function in 'metabolism,' 'intercellular transport,' and 'catabolism' (*Figure 2E*). Categories #2 and #3—which are also highly translated mRNAs but less well expressed at the RNA level than category #1—encode proteins with strikingly different functions: 'development,' 'cell migration,' and 'morphogenesis' (*Figure 2E*). Category #6—which is lowly expressed and modestly translated mRNAs—encode proteins involved in 'signal transduction,' 'differentiation,' and 'development,' including 'nervous system development' (*Figure 2E*). The categories with most *Olfr* genes—#4 and #5—are also only moderately translated (*Figure 2E*). *Upf3b*-null mOSNs had similar numbers of mRNAs in the nine categories as WT mOSNs (compare *Figure 2J* with *Figure 2D*), indicative of UPF3B not altering the mOSN transcriptome and translome globally. Rather, UPF3B influences specific mRNAs, as described above for the mOSN transcriptome, and below for the mOSN translome.

## The relationship between NMD and translation in vivo

NMD is a translation-dependent pathway, based on protein-synthesis inhibitor and transfection experiments in immortalized cell lines (*Belgrader et al., 1993*; *Carter et al., 1995*; *Karousis and*

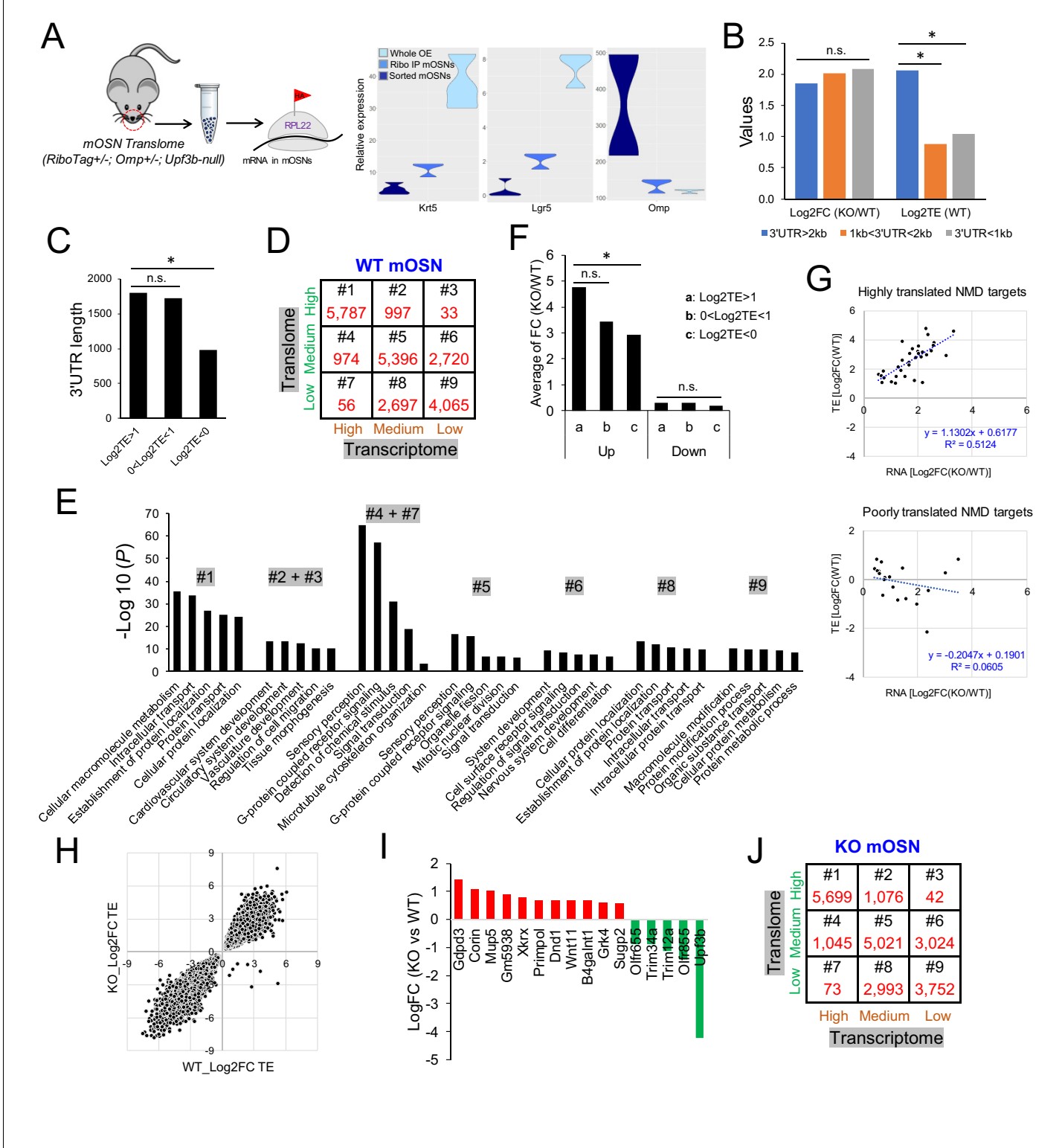

**Figure 2.** The mOSN translome and NMD. (A) Left, strategy used to define the mOSN translome. Right, RNAseq analysis of the expression of gene markers for HBCs (*Krt5*), GBCs (*Lgr5*), and mOSNs (*Omp*) in the indicated samples. (B) Average inferred translation efficiency (TE) of mOSN mRNAs with the indicated 3'UTR length ranges. *, p<0.05. (C) Average 3'UTR length of mOSN mRNAs with the indicated range of TE values. *, p<0.05. (D) mOSN mRNAs from WT mice stratified by steady-state mRNA level (transcriptome) and TE. The number of genes in each category is indicated. (E) Top enriched GO terms associated with the different categories of genes defined in (D). (F) Analysis of upregulated mRNAs (candidate NMD targets) and downregulated mRNAs (indirect targets) are shown on the left and right, respectively. The average shift in expression in *Upf3b*-null mOSNs relative to

*Figure 2 continued on next page*

*Figure 2 continued*

WT mOSNs is shown for mOSNs binned by TE (a and c have the highest and lowest TE values, respectively). *, p<0.05. (G) Scatter plot of the 52 high-confidence mOSN NMD targets, showing TE vs. NMD magnitude (upregulation in *Upf3b*-null mOSNs). Both values are log2-transformed. (H) Scatterplot showing the TE of mRNAs in *Upf3b*-null vs. WT mOSNs. (I) mRNAs exhibiting significantly altered TE in response to *Upf3b* loss. (J) mOSN mRNAs from *Upf3b*-null (KO) mice stratified by steady-state mRNA level (transcriptome) and TE. The number of cells in each category is indicated.

*Mühlemann, 2019*). Our mOSN transcriptome and translome data from *Upf3b*-null and WT mice provided an opportunity to address the relationship of NMD with translation in vivo. Given that higher translation rates allow for a higher frequency of stop codon recognition, it follows that higher translation rates might drive stronger NMD. This predicts that more highly translated mOSN mRNAs will have a higher NMD response than lowly translated mOSN mRNAs. To test this, we binned mRNAs statistically upregulated in *Upf3b*-null mOSNs into three groups stratified by TE. The most highly translated group was statistically more upregulated (i.e., had stronger NMD) than the least translated group (*Figure 2F*, left). As a negative control, we examined downregulated mRNAs (as these would not be direct NMD targets) and found no statistical difference between degree of downregulation and TE (*Figure 2F*, right).

To further examine whether high translation rate is associated with strong NMD magnitude, we binned the 52 high-confidence NMD substrates we defined above into two groups: those with little or no translation and those with high translation (cut-off: log2TE > 1). We then independently plotted these two sets of mRNAs in terms of TE and NMD magnitude (i.e. the degree of upregulation in *Upf3b*-null mOSNs relative to WT mOSNs). The results show that the high-translation group exhibited a correlation between their inferred translation rate and NMD magnitude ($R^2$ = 0.5; *Figure 2G*). In contrast, the low-translation group of mRNAs exhibited no correlation between their translation rate and NMD magnitude ($R^2$ = 0.06; *Figure 2G*). Together, these results support that NMD is translation-dependent in vivo and that its magnitude tends to be enhanced for highly translated mRNAs.

Our mOSN translome data also allowed us to assess the reciprocal question: does *Upf3b* influence translation in vivo? When we plotted the TE of mRNAs when expressed in *Upf3b*-null mOSNs vs. when expressed in WT mOSNs, we found that the vast majority of mRNAs were similarly translated in both genetic backgrounds, as measured by RiboTag analysis (*Figure 2H*, *Supplementary file 2*). Only 16 mOSN mRNAs migrated off the diagonal and thus had a significant change in TE as a result of *Upf3b* loss (*Figure 2H,I*).

## Identification of OE cell clusters

To determine whether UPF3B influences the cellular composition of the OE, we performed scRNA-seq analysis on dissociated OE cells from 4 *Upf3b*-null and 4 WT mice. After filtering out poor quality cells, 25,165 cells remained for subsequent analysis. Biological replicates exhibited similar cell distributions (*Figure 3A*). Using a nonlinear dimensionality-reduction technique—uniform manifold approximation and projection (UMAP)—we identified cell clusters corresponding to 16 known cell types in the OE (*Figure 3B*). Some of the gene markers used to define these cell clusters are shown in *Figure 3C*. Genes exhibiting enriched expression in each of the 16 cell types are listed in *Supplementary file 4*.

Re-clustering of OSN precursors/OSNs (HBCs, GBCs, iOSNs, and mOSNs) revealed several cell sub-clusters within each of these four stages (*Figure 3D,E*). The identification of these sub-clusters suggested that each of these developmental stages exhibit considerable heterogeneity, at least at the transcriptome level. Genes exhibiting enriched expression in each sub-cluster are shown in *Supplementary file 4*.

HBC are known to be reserve stem cells, while GBCs consist of active stem cells and progenitors (*Schwob et al., 2017*). Consistent with this, cell-cycle analysis showed that all four HBC sub-clusters primarily contain quiescent cells, while the GBC sub-clusters have many cells that are proliferating (*Figure 3F*). All four HBC sub-clusters express similar levels of well-established HBC markers, including *Krt5* and *Trp63* (*Figure 4A*). These HBC sub-clusters are each uniquely marked by novel gene markers that we identified (*Figure 4A* and *Supplementary file 4*).

GBCs also segregated into four sub-clusters (*Figure 3D*), which is consistent with past studies demonstrating that GBCs are heterogeneous (*Cau et al., 1997*; *Manglapus et al., 2004*; *Schwob et al., 2017*). Monocle pseudotime trajectory analysis suggested that these four sub-

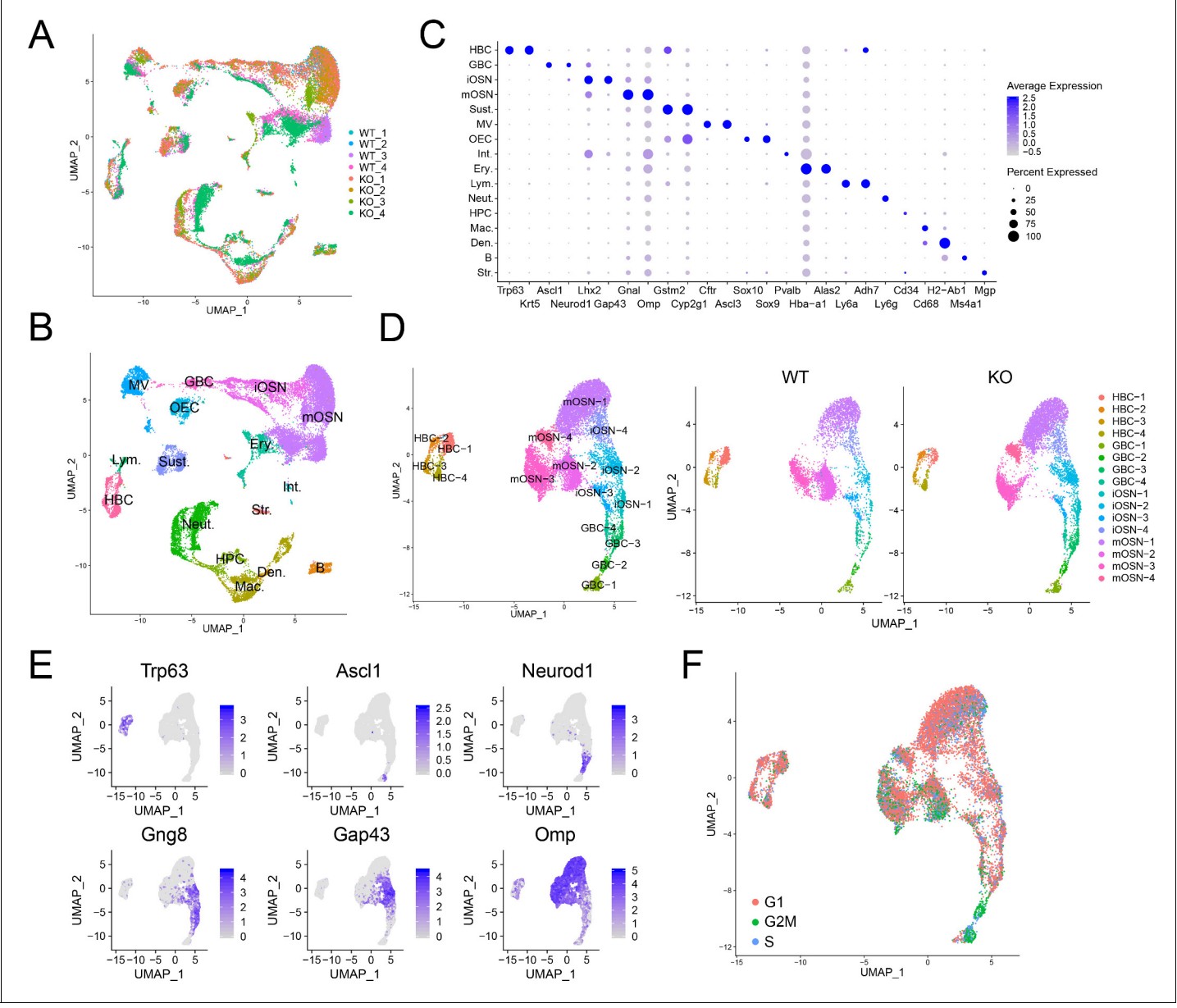

**Figure 3.** Identification of OE cell subsets using scRNAseq analysis. (A) UMAP plot of OE cells from 4 *Upf3b*-null (KO) and 4 WT mice analyzed by scRNAseq. (B) Same UMAP plot as is in (A), showing the identity of the different cell clusters. (C) Dotplot depicting the expression of gene markers in the cell clusters defined in (B). (D) Left, UMAP plot of reclustered OSN precursors/OSNs defined in (A). Right, genotype information. (E) Same UMAP plot as in (D), showing the expression of stage-specific markers. (F) Same UMAP plot as in (D), showing inferred cell-cycle phase based on the expression of a large set of G2/M- and S-phase genes (*Kowalczyk et al., 2015*).

clusters have a linear developmental relationship, with GBC-1 the most immature, GBC-2 more advanced, and GBC-3 and −4 the most advanced (*Figure 4B*). Consistent with this developmental trajectory, both GBC-1 and GBC-2 express the early GBC marker *Ascl1* (*Cau et al., 1997*; *Manglapus et al., 2004*). GBC-1 is likely to be more primitive than GBC-2, based on the frequent and high expression of the later GBC markers *Neurog1* and *Neurod1* (*Cau et al., 1997*; *Manglapus et al., 2004*) in the latter, not the former (*Figure 4C*). While GBC-3 and −4 are clearly GBCs based on the expression of several GBC markers (e.g. *Neurog1* and *Neurod1*), they also express iOSN markers (e.g. *Gng8* and *Gap43* [*Iwema and Schwob, 2003*; *Figure 4C*]), consistent with GBC-3 and -4 being iOSN precursors and hence advanced GBCs. iOSNs also segregated into several cell sub-clusters that each express unique genes (*Figure 4D*). These iOSN sub-clusters follow

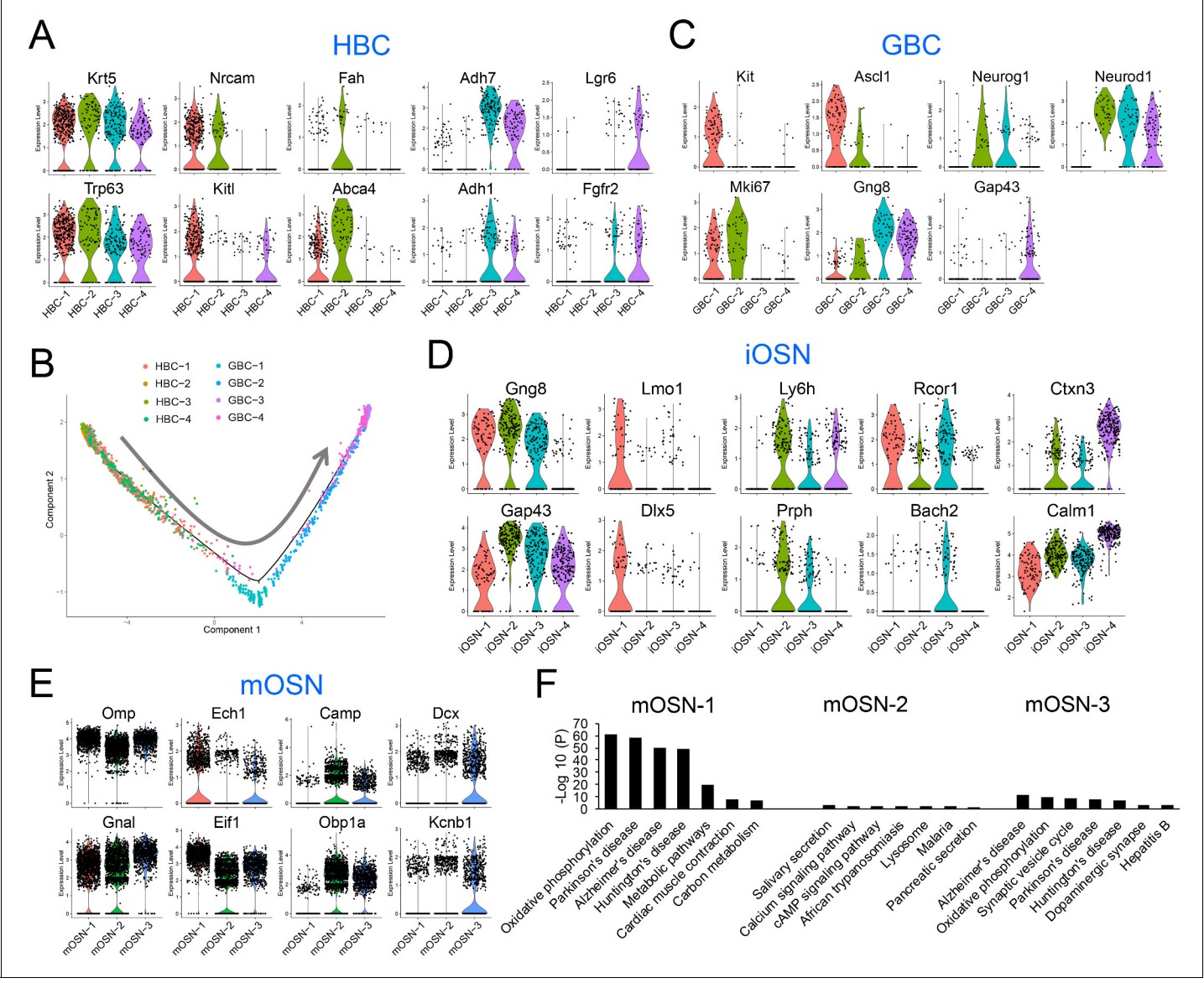

**Figure 4.** HBC, GBC, iOSN and mOSN heterogeneity. (A) Violin plots showing the expression of selective gene markers in the four indicated HBC sub-clusters in WT mice. (B) Monocle trajectory analysis of the HBC and GBC sub-clusters we identified. The arrow indicates the inferred direction of differentiation. (C–E) Violin plots showing the expression of selective gene markers in the indicated GBC, iOSN, and mOSN sub-clusters in WT mice. (F) The most statistically enriched signaling pathways in the mOSN-1,–2, and −3 sub-clusters.

a 'linear' pattern as depicted by the UMAP algorithm (*Figure 3D*), consistent with them representing sequential developmental states, each with unique transcriptomes.

Most WT mOSNs segregated into three different cell clusters (*Figure 3D*), each of which preferentially express different genes (*Figure 4E*). GO and KEGG signaling pathway analyses indicated that these three mOSN sub-clusters are enriched for different functions and signaling pathways, respectively (*Figure 4F*; *Supplementary file 4*).

## OSN molecular pathways

Monocle pseudotime analysis of the OSN precursor/OSN cell clusters indicated that they follow a HBC→GBC→iOSN→mOSN trajectory (*Figure 5A*), consistent with previous studies (*Fletcher et al., 2017*; *Schwob et al., 2017*; *Tepe et al., 2018*). To define candidate molecular events occurring during OSN development, we identified genes whose expression is statistically enriched along this

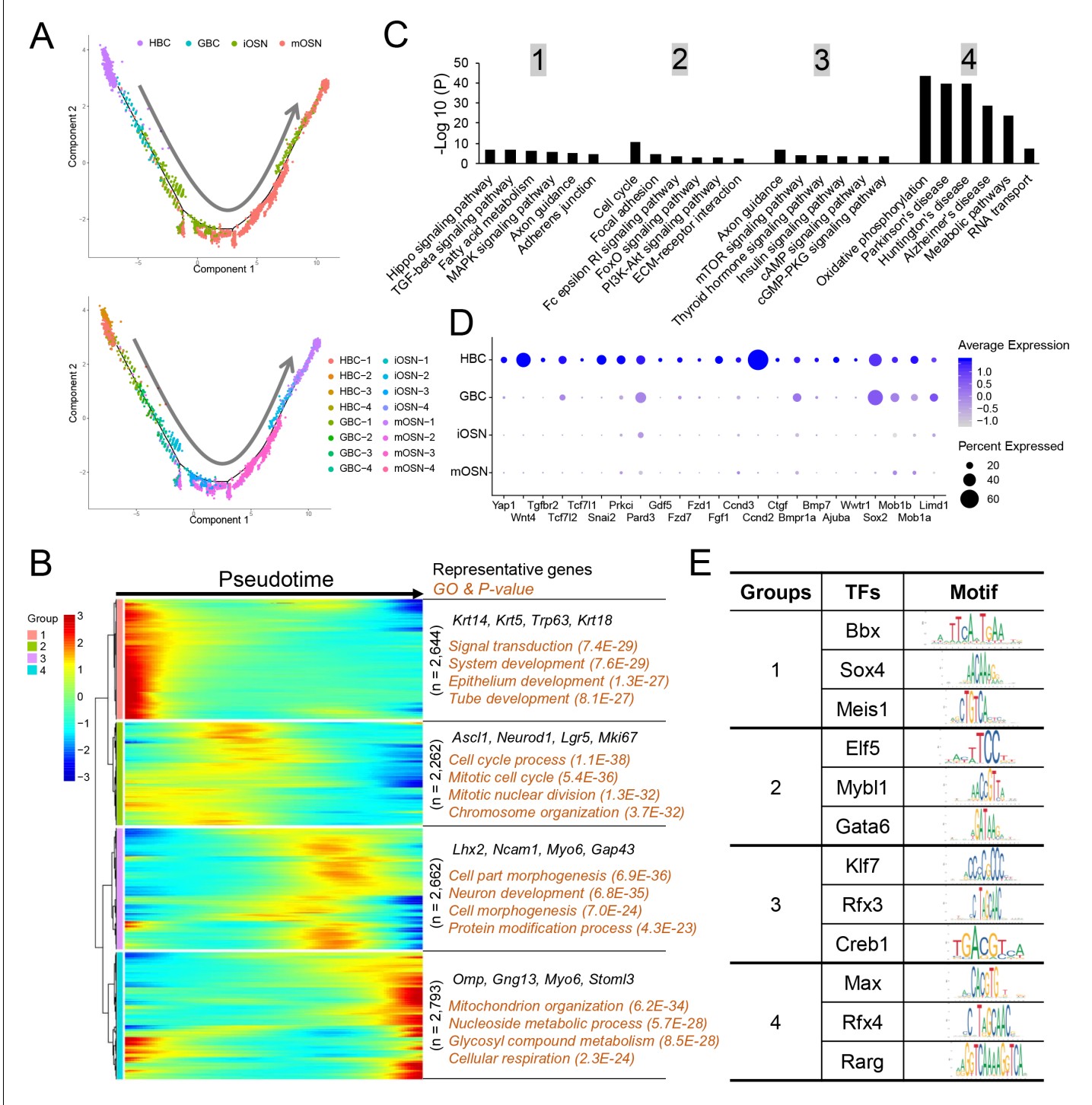

**Figure 5.** Gene groups exhibiting distinct expression dynamics during OSN development. (**A**) Monocle pseudotime trajectory analysis of the indicated cell clusters and sub-clusters from WT mice defined in *Figure 3B* (top) and *Figure 3D* (bottom), respectively. (**B**) Heatmap depicting the expression pattern of the four gene groups we defined, each with a unique expression pattern, as defined by the trajectory timeline shown in (**A**), upper. Top: pseudotime directions; right: the number of differentially expressed genes in each group and representative biological processes and P-values. (**C**) The most statistically enriched signaling pathways corresponding to each of the four gene groups defined in (**B**). (**D**) Dot plot showing genes related to the Hippo signaling pathway are primarily expressed in HBCs. (**E**) Transcription factor genes exhibiting the most statistically enriched expression in each gene group defined in (**B**). Target sequences predicted by the ENCODE database are indicated.

pseudotime trajectory (*Supplementary file 4*). This analysis identified 4 distinct patterns of gene expression dynamics that we named groups 1 to 4 (*Figure 5B*). Group-1 genes are dominated by genes expressed transiently in HBCs, including the previously defined HBC-marker genes *Trp63, Krt5,* and *Krt14*. Group-1 genes are statistically enriched for 'signal transduction' and various 'development' categories (*Figure 5B*). Group-2 genes contain GBC genes; indeed the GBC markers *Ascl1, Neurod1,* and *Lgr5* are enriched in group 2. 'Cell cycle process' is statistically enriched (*Figure 5B*), consistent with the fact that GBCs undergo self-renewal and proliferative expansion. Group-3 genes are mainly expressed in iOSNs, include the well-established iOSN marker genes *Lhx2, Ncam1*, and *Gap43*. 'Neuron development' is enriched in group 3 (*Figure 5B*), consistent with the fact that iOSNs are undergoing the final stages of development prior to becoming mature neurons. Group-4 genes are mainly expressed in mOSNs; enriched GO categories include 'mitochondrion organization,' 'metabolism,' and 'cellular respiration'.

KEGG signaling pathway analysis revealed that genes involved in different signaling pathways are enriched in each of the 4 groups (*Figure 5C*). For example, Hippo pathway genes are enriched in group 1 (*Figure 5D*), raising the possibility this signaling pathway may be important for maintaining HBC stem cells in the quiescent state or eliciting their activation in response to insults.

We also screened for transcription factors preferentially expressed at different stages of OSN development. We identified 209, 178, 169, and 135 transcription factor genes exhibiting enriched expression in groups 1, 2, 3 and 4, respectively (*Supplementary file 4*). The top 3 transcription factors in each group and their DNA-binding specificity are shown in *Figure 5E*.

## UPF3B impacts HBCs and mOSNs

The array of UPF3B-dependent NMD targets we identified in mOSNs (*Figure 2*) raised the possibility that UPF3B has roles in mOSNs and possibly OSN precursors. To assess this, we first determined whether loss of UPF3B impacts the frequency of HBCs, GBCs, iOSNs, and mOSNs. scRNA-seq analysis revealed that there was a significant reduction in the frequency of HBCs in *Upf3b*-null mice relative to WT mice, when compared to either all OSN precursors/OSNs or all OE cells ($p<0.05$; *Figure 6A*). As validation, IHC staining with the HBC marker, TRP63, showed that the density of TRP63+ cells was significantly less in *Upf3b*-null OE than WT OE (*Figure 6—figure supplement 1A*). This effect appeared to be specific, as we observed no significant difference in the relative proportion of GBCs, iOSNs, and mOSNs between *Upf3b*-null and WT mice (*Figure 6—figure supplement 1B*). However, we cannot rule out that the variability among the four samples for each genotype might have obscured a subtle change in the fraction of GBCs, iOSNs, or mOSNs in *Upf3b*-null mice. This variability might either be the result of biological differences between individual mice or differences in dissection and/or cell dissociation. However, as further evidence that the overall frequency of mOSNs was not affected in *Upf3b*-null mice, the mOSN marker, OMP, was similarly expressed (at both the RNA and protein levels) in OE from *Upf3b*-null and WT mice, as assessed in OE preparations obtained from different mice (but of the same genotypes) than those used for scRNA-seq analysis (*Figure 1B* and *Figure 6—figure supplement 1C*).

Our identification of HBC, GBC, iOSN, and mOSN sub-clusters (*Figure 3D*) gave us an opportunity to elucidate whether UPF3B has a role in this unexpected heterogeneity. Despite no significant effect on the mOSN stage as a whole (*Figure 6—figure supplement 1B*), we observed a striking increase in the frequency of 1 of the 4 mOSN sub-clusters—mOSN-4—in *Upf3b*-null mice (*Figure 6B,C*). This sub-cluster represented <1% of all OSNs in most WT mice and increased by an average of 25-fold in *Upf3b*-null mice ($p<0.05$). Conversely, *Upf3b*-null mice had an almost complete loss of another mOSN sub-cluster—mOSN-2—a sub-cluster that was populated by many cells in most WT mice (*Figure 6B,C*). While this reduction failed to reach statistical significance because of variability between samples ($p=0.24$), it is supported by the independent tSNE plots we generated for *Upf3b*-null vs. WT OSNs (*Figure 3D*). Together these results raise the possibility that a 'mOSN subset switch' occurs in *Upf3b*-null mice. Pearson correlation analysis showed that mOSN-2 and −4 sub-clusters are less related to each other in expression profile than they are to the other two mOSN sub-clusters (*Figure 6—figure supplement 2A*). Indeed, these two mOSN sub-clusters have remarkably distinct molecular characteristics (*Figure 6D* and *Supplementary file 4*). Thus, the simultaneous loss and acquisition of these mOSN subsets in *Upf3b*-null mice has the potential to alter olfaction.

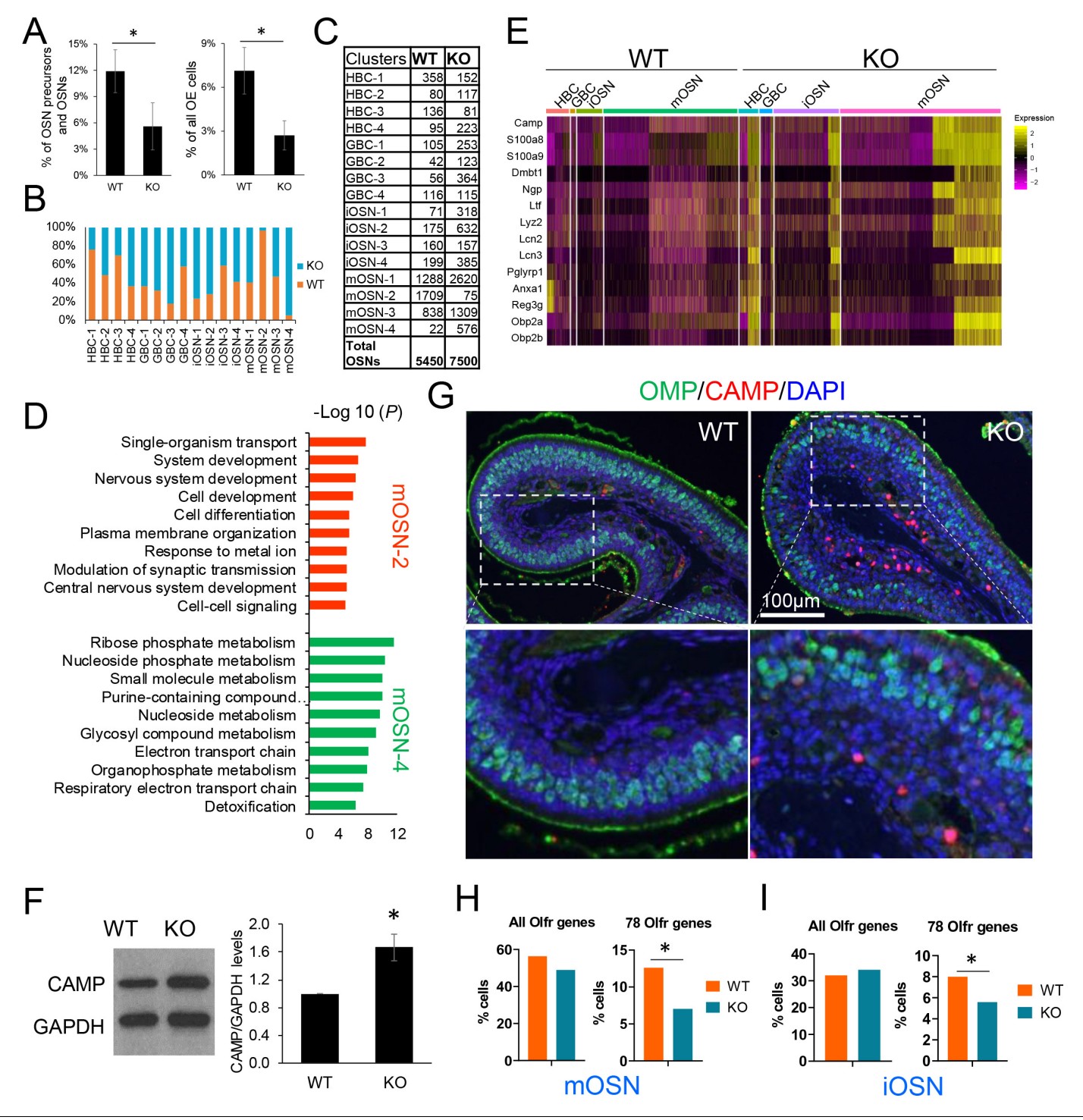

**Figure 6.** UPF3B shapes olfactory neurogenesis. (**A**) The fraction of HBCs per all OSN precursors/OSNs (HBCs, GBCs, iOSNs and mOSNs) (left) or all OE cells (right), in *Upf3b*-null (KO) and WT mice, as determined by scRNA-seq analysis. *, p<0.05. (**B**) The percentage of cells from the indicated cell sub-clusters in *Upf3b*-null (KO) and WT mice, as determined by scRNAseq analysis. (**C**) Cell number in each cell sub-cluster, as defined in *Figure 3D*. (**D**) Most statistically enriched GO terms in the mOSN-2 and −4 sub-clusters. (**E**) Heatmap depicting the expression pattern of anti-microbial genes in the indicated cell subsets. (**F**) Left: Western blot analysis of endogenous CAMP protein level in the OE from *Upf3b*-null (KO) and WT mice. Right: quantification of CAMP level normalized against GAPDH (n = 3). *, p<0.05. (**G**) IF analysis of adult mouse OE sections co-stained with antisera against CAMP (red) and OMP (green). Nuclei were stained with DAPI (blue). (**H, I**) The percentage of mOSNs (**H**) and iOSNs (**I**) in our scRNAseq datasets that

*Figure 6 continued on next page*

*Figure 6 continued*

express *Olfr* genes. Left, all known *Olfr* genes. Right, the 78 *Olfr* genes significantly downregulated in *Upf3b*-null mice, based on RNAseq analysis (*Figure 1C*). *, p<0.05.

The online version of this article includes the following figure supplement(s) for figure 6:

**Figure supplement 1.** Impact of UPF3B loss on OE cell subsets.
**Figure supplement 2.** mOSN subsets and UPF3B-regulated genes.

## UPF3B shapes the OLFR repertoire and suppresses immune gene activation

To define genes that are candidates to act downstream of NMD in different OSN cell populations, we used our scRNA-seq datasets to identify genes differentially expressed in the *Upf3b*-null vs. WT cell clusters (*Supplementary file 4*). This revealed that a major category of *Upf3b*-regulated genes in OSNs are immune genes, including a large fraction of genes encoding antimicrobial proteins (*Supplementary file 5*). This was intriguing, as it raised the possibility that OSNs not only normally function in olfaction but also in defense against microbes, a reasonable possibility given that the OE is direct contact with the outside environment. The expression of anti-microbial genes was not confined to the mature neurons in the OE (i.e. mOSNs), as we found that most of these immune-defense genes were also expressed and upregulated in *Upf3b*-null mice at the HBC, GBC, and iOSNs stages (*Figure 6E*). More than half (48 out of 88) of these upregulated mRNAs encoding immune-related proteins harbor at least one NIF (*Supplementary file 5*). This suggests that many of these mRNAs encoding immune system factors are directly targeted for decay by the NMD pathway. This has interesting potential physiological consequences, as described in the Discussion.

Among the antimicrobial genes expressed and upregulated in *Upf3b*-null OSN precursors and OSNs was *Camp* (also known as '*Cramp*'), which encodes a member of the cathelicidin family of anti-microbial peptides that has an important role in the defense against microbial infections, and functions in cell chemotaxis, immune mediator induction, and inflammatory response regulation (*Zhang and Gallo, 2016*). To further assess its regulation, we performed western blot analysis with a validated anti-CAMP antiserum, which showed that CAMP protein is expressed in the OE and is upregulated in *Upf3b*-null mice (*Figure 6F*). As further evidence, immunofluorescence analysis detected modest anti-CAMP staining in OE cells (as well as strong staining in the in the lamina propria), both of which were increased in *Upf3b*-null mice (*Figure 6G* and *Figure 6—figure supplement 2B*).

The other major category of genes that we discovered are regulated by *Upf3b* in the OE is *Olfr* genes. As described above, our RNA-seq analysis of purified mOSNs from *Upf3b*-null and WT mice revealed that the *majority* of genes downregulated in response to *Upf3b* loss are *Olfr* genes (*Figure 1C*). In total, we identified 78 *Olfr* genes significantly downregulated in *Upf3b*-null mOSNs (*Figure 1C*). We considered the possibility that these 78 *Olfr* genes are regulated by *Upf3b* through a common local *cis*-acting regulatory sequence, but against this hypothesis, we found that these 78 *Olfr* genes are widely chromosomally distributed (*Figure 6—figure supplement 2C*).

There are more than 1000 olfactory receptor (*Olfr*) genes in mice (*Zhang and Firestein, 2002*). Individual OSNs select a single *Olfr* gene for expression from this large repertoire (*Chess et al., 1994*; *Malnic et al., 1999*; *Serizawa et al., 2003*). This unique mechanism raised the possibility that rather than regulating *Olfr* expression per se, *Upf3b* might instead influence the decision whether or not specific *Olfr* genes are selected to be the dominant receptors in mOSNs. In other words, *Upf3b* might increase the probability that these 78 *Olfr* genes that are selected to be expressed in individual mOSNs. If selected less often in *Upf3b*-null OSNs, these 78 *Olfr* genes would appear to be downregulated, but would instead be expressed in fewer OSN. To address this model, we made use of our scRNA-seq datasets. We found that 490 of 3887 mOSNs in WT mice (13%) express one of these 78 *Olfr* genes as the dominant *Olfr* gene. In contrast, only 328 of 4654 mOSNs in *Upf3b*-null mice (7%) express one of these 78 *Olfr* genes as the dominant *Olfr* gene (*Figure 6H*). In contrast, when all known *Olfr* genes were considered as a group, there was no significant difference in the percentage of *Olfr* genes selected in *Upf3b*-null vs. WT mOSNs (*Figure 6H*). Likewise, we found that these 78 *Olfr* genes were under-represented in *Upf3b*-null iOSNs as compared with WT iOSNs (5.6% vs 8.0%, p<0.05) (*Figure 6I*). Together, these results support a model in which UPF3B promotes the

selection of these 78 *Olfr* genes to be the dominant *Olfr* gene expressed in individual mOSNs, a model we elaborate on in the Discussion.

## Discussion

NMD factors have been shown to have numerous roles in the development and function of neurons (*Jaffrey and Wilkinson, 2018*). As described in the Introduction, the NMD factor examined in our study – UPF3B – has been shown to be necessary for normal cognition in humans and its loss is associated with several neuro-developmental disorders (*Nguyen et al., 2014*; *Tarpey et al., 2007*). While the precise roles of UPF3B in behaviors is not known, it has been shown that UPF3B is critical for both neural differentiation and mature neuronal functions (*Alrahbeni et al., 2015*; *Huang et al., 2018*; *Jolly et al., 2013*). In addition to *UPF3B*, other NMD genes are likely to have roles in the nervous system (*Jaffrey and Wilkinson, 2018*). For example, copy-number gain and loss of several genes encoding proteins involved in NMD—including *UPF2, UPF3A, SMG6, RBM8A, EIF4A3,* and *RNPS1*—are statistically significantly associated with neural-developmental disorders in humans (*Nguyen et al., 2013*). Mutations in *RBM8A* have been shown to cause TAR syndrome, which can cause cognitive dysfunction (*Jaffrey and Wilkinson, 2018*). In mice, loss of a single copy of *Rbm8a* or other EJC genes (*Magoh* or *Eif4e*) causes microcephaly and severe neural defects (*Mao et al., 2017*). In worms, flies and mice, genetic perturbation of other NMD genes causes neural defects, including synaptic and axon guidance defects (*Colak et al., 2013*; *Giorgi et al., 2007*; *Long et al., 2010*; *Zheng et al., 2012*). Two recent studies revealed that conditional loss of the NMD gene, *Upf2*, in specific neural populations in mice causes a variety of intriguing defects, including aberrant behavior, spine density, and synaptic plasticity (*Johnson et al., 2019*; *Notaras et al., 2019*). Together, these studies make a strong case that NMD has roles in the CNS.

Here, we report the first investigation of the role of NMD in the olfactory system. One of our major findings was that *Upf3b* loss causes shifts in gene expression in OSNs. One major class of genes impacted by *Upf3b* is the *Olfr* genes. These genes have evolved to allow recognition of the large array of odors encountered by higher organisms. In mice, there are >1000 *Olfr* genes, each of which encode a G-coupled receptor that binds to a restricted set of odorants (*Godfrey et al., 2004*; *Zhang and Firestein, 2002*). In order to interpret the information from a given odor, it is critical that only a single OLFR be expressed in each mOSN. This is accomplished by a novel gene regulatory mechanism that selects only a single *Olfr* gene to be expressed in any given mOSN (*Chess et al., 1994*; *Malnic et al., 1999*; *Serizawa et al., 2003*). While the underlying mechanism for this 'one-neuron-one-receptor' rule is not fully understood, a prevailing model is that a stochastic mechanism drives a single *Olfr* to become dominate transcriptionally, a decision that is reinforced by feedback mechanisms (*Dalton et al., 2013*; *Lewcock and Reed, 2004*; *Serizawa et al., 2004*; *Serizawa et al., 2005*).

The first indication that UPF3B might have a role in the selection of *Olfr* genes came from our RNA-seq analysis, which revealed that the *majority* of genes expressed at lower level in *Upf3b*-null mOSNs are *Olfr* genes. In total, we found that 78 *Olfr* genes are statistically downregulated in *Upf3b*-null mOSN. To address mechanism, we performed scRNA-seq analysis and found that these 78 *Olfr* genes are rarely represented as the dominant genes in individual mOSNs in *Upf3b*-null mice. This defect was also present at the iOSN stage, suggesting that *Upf3b* is involved directly or indirectly in determining which *Olfr* gene are selected for dominant expression during OSN development.

A caveat is the OE contains zones enriched for mOSNs expressing particular sets of OLFRs (*Miyamichi et al., 2005*; *Ressler et al., 1994*), and thus even though we made an effort to dissect the entire OE for RNA-seq analysis, it is possible that there is zonal heterogeneity in the samples we analyzed. To reduce this potential bias, we pooled dissociated OE cells from 3 mice for FACS sorting. Confidence that the 78 *Olfr* genes are regulated by *Upf3b* comes from the reproducibility of the regulation in independent samples (*Figure 1C*) and validation by qPCR (*Figure 1—figure supplement 2B*). Furthermore, our single-cell RNA-seq analysis (which analyzed samples different from those analyzed by RNA-seq) verified the regulation of these 78 *Olfr* genes (*Figure 6H*).

How might *Upf3b* influence the selection of this particular set of *Olfr* genes? Given that UPF3B is a NMD factor, it could promote the decay of an mRNA encoding a repressor that acts to regulate the selection of these 78 *Olfr* genes for dominant expression. To test this model, we screened genes

exhibiting significantly upregulated expression in *Upf3b*-null OSNs for those that encode factors known to regulate *Olfr* gene expression or have binding sites in *Olfr* promoters (*Clowney et al., 2011*; *Dalton et al., 2013*; *Hirota and Mombaerts, 2004*; *Markenscoff-Papadimitriou et al., 2014*; *McIntyre et al., 2008*; *Michaloski et al., 2006*; *Wang et al., 1997*). This screen identified two genes—*Mafg* and *Irf8*—that fulfilled this criteria. Both encode transcriptional repressors (*Igarashi et al., 1994*; *Salem et al., 2014*) that bind O/E consensus sites found in *Olfr* gene promoters (*Michaloski et al., 2006*). Thus, *Mafg* and *Irf8* are candidates to act directly downstream of NMD in a regulatory circuit that suppresses the transcription of these 78 Olfr genes. *Mafg* is a member of the Maf subfamily of basic leucine-zipper transcription factor genes that encode small proteins containing a B-ZAP DNA-binding domain, but lack a transactivation domain, and thus members of this family dimerize to form transcriptional repressors (*Igarashi et al., 1994*). MAFG is best known for its ability to regulate globin transcription in erythroid cells; our results raise the possibility that MAFG also functions in OSNs to regulates *Olfr* genes. IRF8 regulates the development hematopoietic cells; its expression in OSNs raises the possibility that this transcription factor also functions in OSNs.

Our findings support a model in which IRF8 and MAFG normally subtly repress the transcription of a subset of *Olfr* genes in OSNs to fine-tune their expression. Our evidence suggests that IRF8 and MAFG are encoded by NMD target mRNAs, so when NMD is disrupted, these transcriptional repressors are overexpressed, leading to reduced expression of their *Olfr* gene targets in developing OSNs. Thus, NMD deficiency would be expected to reduce the probability that these particular *Olfr* genes will be chosen to be the 'dominant *Olfr* gene' in individual mOSNs, which is precisely what we observed in *Upf3b*-null mice.

A non-mutually exclusive possibility is that *Upf3b* dictates the selection of *Olfr* genes by influencing OSN development. In support, several of the genes we found were regulated by *Upf3b* have been reported to play essential roles in neurogenesis, including *Lrp2*, *Hk2*, *Notch2*, *Gdf11*, *Fos*, *Ptch1*, *Spry2*, and *Cwc22*. *Upf3b* could also indirectly influence the *Olfr* repertoire by differentially affecting the survival of OSNs harboring different OLFRs. In support, we found that *Upf3b* loss upregulates *Fos*, which is associated with OSN apoptosis (*Michel et al., 1994*).

The other major class of genes regulated by *Upf3b* in OSNs is antimicrobial genes. This finding, coupled with our finding that OSNs constitutively express these anti-microbial genes (albeit at low levels), suggests that OSNs function not only in olfaction but also in defense against microbes in the bronchial airways. In support, a recent study showed that inflammation causes OSNs to switch from a role in olfaction to immune defense (*Chen et al., 2019*). This raises the interesting possibility that loss of *Upf3b* triggers OE inflammation, which, in turn, diverts OSNs from functioning in olfaction to immune defense, thereby causing deficient olfaction. In support, another recent study reported that NMD disruption causes neuro-inflammation in the central nervous system (*Johnson et al., 2019*). In particular, this study found that *Upf2* conditional knockout in the murine forebrain leads to immune infiltration, coupled with deficits in memory, synaptic plasticity, social, and vocal communication (*Johnson et al., 2019*). Importantly, they found that anti-inflammatory agents partially rescued many of these deficits, indicating that the inflammation is at least partially responsible for the neural defects in these *Upf2*-conditional knockout mice. It will be intriguing to determine whether humans with *UPF3B* mutations also suffer from neuro-inflammation and whether this is responsible for their intellectual disability.

Our finding that loss of UPF3B upregulates a very large number of immune-related genes in OSNs, over half of which encode mRNAs that have NIFs and thus may be direct NMD targets (*Supplementary file 5*), raises the possibility that this 'immune induction' response to NMD inhibition is physiologically important. In this regard, it is notable that some viruses have been shown to inhibit NMD, and, in turn, NMD can inhibit viral infection (*Wachter and Hartmann, 2014*; *Wada et al., 2018*). Coupled with our data, these findings raise the intriguing possibility that the reason that OSNs express high levels of antimicrobial genes in response to NMD inhibition is because this provides a means to cope with infectious agents, particularly those that inhibit NMD as a means to avoid the antiviral actions of NMD.

In addition to NMD inhibition directly upregulating mRNAs encoding immune factors in OSNs, we identified candidate intermediary factors that may act in a circuit to achieve the same aim. In particular, we identified three mRNAs—*Notch2*, *Bhlhe40*, and *Rac2*—which are high-confidence NMD targets in mOSNs (*Table 1*) that encode factors previously shown to regulate the expression of

many genes encoding inflammatory mediators and antimicrobial proteins (*Dooley et al., 2009*; *Jarjour et al., 2019*; *Shang et al., 2016*).

Our scRNA-seq analysis indicated that *Upf3b* impacts the steady-state frequency of specific OSN precursor and OSN cell subsets. We found that *Upf3b*-null mice have decreased numbers of HBCs, suggesting that UPF3B promotes the maintenance of these reserve stem cells. This effect appeared to be specific, as we observed no significant effects on GBCs, which also serve as olfactory stem cells, but unlike HBCs, function to generate new mOSNs constitutively (*Schwob et al., 2017*). We also observed that *Upf3b*-null mice acquired a specific group of mOSNs harboring a unique transcriptome that are hardly present in WT mice. This mOSN-4 sub-cluster is enriched for many genes, such as *Tuba1a, Nsg1, Chchd10, Eml2, Ubb,* and *Gldc*, which suggests that *Upf3b* normally represses these genes. It remains to be determined whether the aberrant over-expression of these genes causes aberrant mOSN function. We also found that *Upf3b*-null mice largely lack a mOSN sub-cluster—mOSN-2—that we found contained large numbers of cells in most WT mice. The mOSN-2 sub-cluster is likely to be functional, as genes enriched in this sub-cluster include *Pten, App, Cnga2, Nrp2, Ncam1, Adcy3, Gnal, Atf5*, and *Gfy* (*Supplementary file 4*), all of which are known to be essential for olfactory epithelium development and/or olfaction. This reciprocal shift in these two mOSN sub-clusters in *Upf3b*-null mice raises the possibility that UPF3B loss converts the mOSN-2 sub-cluster into the mOSN-4 sub-cluster. This remains to be determined, as does the physiological consequences of these shifts in mOSN sub-populations. Another important area for future investigation is to determine whether these cell-subset alterations in *Upf3b*-null mice are cell autonomous or non-cell autonomous.

As described in the Introduction, few direct NMD target RNAs have previously been defined in vivo. Our study fills this gap by identifying high-confidence NMD target mRNAs in mOSNs in vivo. Many of the NMD targets we identified in mOSNs have long 3'UTRs, raising the possibility that mOSNs have a predilection for degrading mRNAs with this particular NIF. By analogy, evidence suggests that mRNAs harboring long 3'UTRs are also preferentially targeted for destruction by NMD in male germ cells (*Bao et al., 2016*). Several of the NMD target mRNAs that we identified in mOSNs are good candidates to have roles in OSN development. For example, *Gdf11* functions in negative-feedback control of OE neurogenesis; *Lrp2* promotes the proliferation of neural precursor cells in the subependymal zone of the olfactory bulb; and *Notch2* is required for maintaining sustentacular cell function in the OE (*Gajera et al., 2010*; *Kawauchi et al., 2009*; *Rodriguez et al., 2008*). Other NMD target mRNAs that we identified, including *Ptch1* and *Hk2*, encode proteins known to be important for the development of neurons outside of the olfactory system (*Iulianella and Stanton-Turcotte, 2019*; *Zheng et al., 2016*).

Our study provides a useful resource for the olfactory field. For example, our scRNA-seq analysis identified putative new OSN precursor and OSN cell subsets. While we do not know the significance of this heterogeneity, the genes differentially expressed by the sub-clusters we identified suggests functional relevance. For example, the genes differentially expressed by the 4 cell-sub clusters we identified for both GBCs and iOSNs suggested that these sub-clusters represent distinct developmental stages. Our results are consistent with Fletcher et al., who demonstrated that that the 1 GBC and 4 INP/iOSN sub-clusters they identified follow a linear developmental pattern (*Fletcher et al., 2017*). Our genome-wide determination of mOSN mRNA expression levels and ribosome occupancy (i.e. translation rates) will be useful for future studies to determine how transcription, translation, and other post-transcriptional processes coordinate to regulate the expression of large sets of genes in mature neurons in vivo. We divided mOSN-expressed mRNAs into nine categories based on steady-state mRNA level and ribosome occupancy, allowing dissection of common functions encoded by similarly regulated mRNAs. Given that translation is a highly energy-consuming process (*Lynch and Marinov, 2015*), it is likely that there has been strong selection pressure for many mRNAs to be translated inefficiently. Indeed, we found that modestly translated mRNAs encode many key mOSN proteins, including receptors, signaling factors, and developmental regulators.

In conclusion, our study provides an invaluable set of resources for the olfactory field and identifies a post-transcriptional regulatory pathway that impacts OSNs.

# Materials and methods

**Key resources table**

| Reagent type (species) or resource | Designation | Source or reference | Identifiers | Additional information |
|---|---|---|---|---|
| Gene (*Mus musculus*) | *Upf3b* | GenBank | Gene ID: 68134 | |
| Genetic reagent (*Mus. musculus*) | C57BL/6J | Jackson Laboratory | Stock #: 000664 RRID:MGI:3028467 | |
| Genetic reagent (*Mus. musculus*) | *Upf3b*-null mice | PMID:21925383 | RRID:MGI:6110148 | Miles Wilkinson lab |
| Genetic reagent (*Mus. musculus*) | *R26-eYFP* mice | PMID:11299042 | | Obtained from Dr. Maike Sander (UCSD) |
| Genetic reagent (*Mus. musculus*) | *Omp-Cre* mice | PMID:22057188 | | Obtained from Dr. Haiqing Zhao (Johns Hopkins University) |
| Genetic reagent (*Mus. musculus*) | *RiboTag* mice | PMID:19666516 | | Obtained from Dr. Paul Ameiux (University of Washington) |
| Antibody | Rabbit monoclonal anti-OMP (EPR19190) | Abcam | Cat# ab183947 RRID:AB_2858281 | IF (1:400), WB (1:2000) |
| Antibody | Goat polyclonal anti-OMP | FUJIFILM Wako Chemicals | Cat# 544–10001-WAKO RRID:AB_2315007 | IF (1:200) |
| Antibody | Rabbit polyclonal anti-CAMP | Generated by Richard L. Gallo laboratory | PMID:11442754 | IF (1:200) |
| Antibody | Rabbit polyclonal anti-FUT10 | Proteintech | Cat#: 18660–1-AP RRID:AB_10641997 | IF (1:200) |
| Antibody | Donkey anti-Goat IgG (H+L) Cross-Adsorbed Secondary Antibody, Alexa Fluor 488 | Thermo Fisher Scientific | Cat#: A-11055 RRID:AB_2534102 | IF (1:1000) |
| Antibody | Donkey anti-Rabbit IgG (H+L) Highly Cross-Adsorbed Secondary Antibody, Alexa Fluor 555 | Thermo Fisher Scientific | Cat#: A-31572 RRID:AB_162543 | IF (1:1000) |
| Sequence-based reagent | Fosl2_F | This paper | PCR primers | CCGCAGAAGGAGAGATGAG (from IDT) |
| Sequenced-based reagent | Fosl2_R | This paper | PCR primers | GCAGCTTCTCTGTCAGCTC (from IDT) |
| Sequence-based reagent | Ptger2_F | This paper | PCR primers | TGCTCCTTGCCTTTCACAATC (from IDT) |
| Sequenced-based reagent | Ptger2_R | This paper | PCR primers | CCTAAGTATGGCAAAGACCCAAG (from IDT) |
| Sequence-based reagent | Adcy6_F | This paper | PCR primers | TTCCTGACCGTGCCTTCTC (from IDT) |
| Sequenced-based reagent | Adcy6_R | This paper | PCR primers | CACCCCGGTTGTCTTTGC (from IDT) |
| Sequence-based reagent | Ptch1_F | This paper | PCR primers | ACCTCCTAGGTAAGCCTCC (from IDT) |
| Sequenced-based reagent | Ptch1_R | This paper | PCR primers | CACCCACAATCAACTCCTCC (from IDT) |
| Sequence-based reagent | Cwc22_F | This paper | PCR primers | CAGAAGACAGATACACAGAGCAAG (from IDT) |

*Continued on next page*

*Continued*

| Reagent type (species) or resource | Designation | Source or reference | Identifiers | Additional information |
|---|---|---|---|---|
| Sequenced-based reagent | Cwc22_R | This paper | PCR primers | CTCTCTCTCTCTCTCTGCGTTT (from IDT) |
| Sequence-based reagent | Fut10_F | This paper | PCR primers | CCAGGGCCTTCCTATTCTACG (from IDT) |
| Sequenced-based reagent | Fut10_R | This paper | PCR primers | CTGAATGTGGCCGTATGGTTG (from IDT) |
| Sequence-based reagent | Gdpd3_F | This paper | PCR primers | TGATCCGACACTTGCAGGAC (from IDT) |
| Sequenced-based reagent | Gdpd3_R | This paper | PCR primers | GCTGTGGGGTAATCGGTCAT (from IDT) |
| Sequence-based reagent | Olfr827_F | This paper | PCR primers | TGGGATGGTTCTTCTGGGAA (from IDT) |
| Sequenced-based reagent | Olfr827_R | This paper | PCR primers | ACCGTGGAGTAGGAGAGGTC (from IDT) |
| Sequence-based reagent | Rpl19_F | This paper | PCR primers | CCTGAAGGTCAAAGGGAATGTG (from IDT) |
| Sequence-based reagent | Rpl19_R | This paper | PCR primers | CTTTCGTGCTTCCTTGGTCTT (from IDT) |
| Commercial assay or kit | Chromium Single Cell 3' Library and Gel Bead Kit | 10X Genomics | Cat# 120237 | |
| Commercial assay or kit | iScript cDNA synthesis Kit | BioRad | Cat# 170–8891 | |
| Commercial assay or kit | SsoAdvanceD Universal SYBR Green Supermix | BioRad | Cat# 172–5274 | |
| Commercial assay or kit | RNeasy Mini Kit | Qiagen | Cat# 74104 | |
| Software, algorithm | Cell Ranger Version 2.1.1 | 10x genomics | Cell Ranger Version 2.1.1 | |
| Software, algorithm | Seurat (v3.1.5) | Designed by Rahul Satija laboratory | PMID:31178118 | |
| Software, algorithm | Monocle (v2.16.0) | Designed by Cole Trapnell laboratory | PMID:28114287 | |
| Software, algorithm | NIH ImageJ (v1.8.0) | NIH | Version 1.8.0 | |

## Mice

This study was carried out in strict accordance with the Guidelines of the Institutional Animal Care and Use Committee (IACUC) at the University of California, San Diego. The protocol was approved by the IACUC at the University of California, San Diego (permit number: S09160). All studies were conducted on adult male mice housed under a 12 hr light:12 hr dark cycle and provided with food and water ad libitum. Of note, we only performed analyses on male mice. Since *Upf3b* is X-linked gene, we analyzed Upf3b$^{+/y}$ (WT) and Upf3$^{-/y}$ (KO) mice. All mouse strains used for analysis were backcrossed to C57BL/6J for at least eight passages.

## Behavioral and weight analyses

To assess the effect of UPF3B loss on mouse weight, 19male pups (nine *Upf3b*-null and ten WT mice) from *Upf3b*$^{+/-}$ × WT breeders (6 litters) were assessed, performed as described previously (*Tan et al., 2016*). For pre-weaning pups, to reduce stress, forceps and gloves were changed frequently between cages.

For the coyote/bobcat urine experiment, 10 male mice (10- to 16 weeks of age) from each genotype were analyzed. Each mouse was placed into a cage for 10 min to acclimatize, a strip of filter

paper soaked with coyote urine (Snow Joe) or bobcat urine (Predator Pee) was placed into the cage for 5 min, and the amount of time the mouse was in the vicinity of the filter paper was determined by video recording. Each mouse was tested separately in the absence of humans or other mice in the room.

## RNA-seq analysis

For each mOSN sample analyzed, 3 C57BL/6J male mice (8- to 9-weeks old) were pooled. Four replicate samples were analyzed per genotype (*Upf3b$^{+/y}$; Omp-Cre; R26-eYFP* and *Upf3b$^{-/y}$; Omp-Cre; R26-eYFP*). Cell sorting experiments were performed on two separate days, with two samples sorted per day. The OE was dissected as described (*Gong, 2012*) and dissociated using the Papain Dissociation System (Worthington) at 37°C for 15 min, followed by extensive trituration. Cells were filtered using a 40-μm strainer (Falcon). After spinning at 200 *g* for 5 min, cells were resuspended in Hanks' balanced salt solution (HBSS) containing 3% FBS (Gibco) but without Ca$^{2+}$ and Mg$^{2+}$. The cell suspension was mixed with propidium iodide (final concentration of 1 μg/ml) and the OMP-eYFP$^+$ cells were sorted by flow-cytometry. RNA was isolated from the OMP-eYFP$^+$ cells using Tri-ZOL (Life Technologies), followed by a secondary purification step using a RNeasy column (Qiagen). Total RNA was assessed for quality using an Agilent Bioanalyzer, and samples determined to have an RNA Integrity Number (RIN) of at least 8 or greater were used to generate RNA libraries using Illumina's TruSeq RNA Sample Prep Kit, following the manufacturer's specifications, with the RNA fragmentation time adjusted to 5 min. RNA-seq was performed at the Institute of Genomic Medicine at UCSD. RNA libraries were multiplexed and sequenced with 100 base pair (bp) pair end reads on an Illumina HiSeq4000. The average number of reads per sample ranged from approximately 15 to 22 million reads. Reads were filtered for quality and aligned with STAR (2.5.2b) against *Mus musculus* release-90, Ensembl genome (GRCm38). The exon counts were aggregated for each gene to build a read count table using SubRead function featureCounts (*Liao et al., 2014*). Using the exon start/end positions, we extracted the exon sequences from the mm10 mouse genome, and ligated them together in silico for each transcript. For each entry, the entire transcript sequence was subtracted from the known CDS sequence (obtained as above) to identify 3'UTR length. DEGs were defined using DESeq2 (*Love et al., 2014*) using a threshold *q*-val of <0.05. The R package program 'pheatmap' was used for clustering and to generate heatmap plots. GO analysis was done using database for annotation, visualization and integrated discovery (DAVID), version v6.8. To infer relative RNA stability, we used the REMBRANDTS program (*Alkallas et al., 2017*) following the tutorial (https://github.com/csglab/REMBRANDTS).

## RiboTag analysis

For each mOSN sample analyzed, three C57BL/6J male mice (8 to 9-weeks old) were pooled. Three replicate samples were analyzed per genotype (*Upf3b$^{+/y}$; Omp-Cre; RiboTag* and *Upf3b$^{-/y}$; Omp-Cre; RiboTag*). The OEs was dissected as described (*Gong, 2012*), homogenized, washed with HBSS, centrifuged at 16,000 *g* at 4°C for 10 min, the supernatant was transferred into a new tube and incubated with HA antisera (#16B12; BioLegend, CA) at 4°C for 2.5 hr. Ribosome-bound RNAs were captured on anti-HA agarose beads (Pierce) for 1 hr at 4°C on a tube rotator. RNA libraries were multiplexed and sequenced with 50 bp single-end reads on an Illumina HiSeq4000. RNA sequencing, alignment, and downstream analyses were done as described above for RNA-seq analysis. TE was determined by dividing RiboTag reads by RNA-seq reads. Log2-transformed transcripts per million (TPM) values were used to segregate mRNAs into different categories.

## scRNA-seq analysis

Four C57BL/6J male mice (7 to 8-weeks old) per genotype (*Upf3b$^{+/y}$* and *Upf3b$^{-/y}$*) were used to obtain OE for scRNA-seq analysis. After dissecting the OE as described (*Gong, 2012*), the cells were dissociated following the 10X Genomics Chromium sample preparation protocol. Briefly, tissue was cut into 1 mm$^3$ pieces and digested in HBSS without Ca$^{2+}$ and Mg$^{2+}$ and supplemented with 44 U/ml Dispase (Invitrogen), 1000 U/ml Collagenase type II (Invitrogen) and 10 mg/ml DNaseI (Sigma), for 20 min at 37°C with gentle agitation. The digested tissue was centrifuged at 300 rcf for 5 min and washed in HBSS without Ca$^{2+}$ and Mg$^{2+}$. Dissociated cells were resuspended in 3% FBS in PBS. Dead cells were removed using the ClioCell Dead Cell Removal kit (Amsbio) following the

manufacturer's instructions. Single cells were resuspended in 0.04% BSA in PBS (w/v) and loaded on the 10x Chromium chip. Cell capturing, and library preparation was carried as per kit instructions (Chromium Single Cell Kit [v2 chemistry]). The resultant libraries were size selected, pooled, and sequenced using $2 \times 100$ paired-end sequencing protocol on an Illumina HiSeq 4000 instrument. The libraries initially underwent shallow sequencing to access quality and to adjust subsequent sequencing depth based on the capture rate and unique molecular indices (UMI) detected. All sequencing analyses were performed at the Institute of Genomic Medicine at UCSD.

As described previously (*Sohni et al., 2019*; *Tan et al., 2020b*; *Tan et al., 2020a*), de-multiplexed raw sequencing reads were processed and mapped to the mouse genome (mm10) using Cell Ranger software (v2.0) with default parameters. We filtered raw count matrices by excluding cells expressing less than 200 detectably expressed genes and genes expressed in less than 3 cells. Each library was tagged with a library batch ID and combined across independent experiments using the Seurat package (*Butler et al., 2018*) in R. To check the quality of the single-cell data and to remove multiplets, we performed Seurat-based filtering of cells based on three criteria: number of detected features (nFeature_RNA) per cell, number of UMIs expressed per cell (nCount_RNA) and mitochondrial content, using the following threshold parameters: nFeature_RNA (>500), nCount_RNA (>1,500), and percentage of mitochondrial genes expressed (<0.2%). We used known lineage marker profiles to exclude cell multiplets (cells expressing different lineage markers) and cell-free droplets. Gene expression values were log normalized and regressed by mitochondrial expression ('percent.mt') and cell cycle gene expession ('S.Score' and 'G2M.Score') using the *SCTransform* function. Batch correction was performed using the *JackStraw* functions in the Seurat package.

To identify cell clusters, we employed the UMAP algorithm (*Becht et al., 2019*). The *FindMarkers* function (a Wilcoxon rank sum test) was used to determine differential gene expression between clusters (set at minimum expression in 25% of cells). The *DoHeatmap* function was used to generate an expression heatmap for specific cells and features. GO analysis (DAVID v6.8) was done using the top differentially (positively) expressed genes, with a p-adjusted cut off of 0.01.

Single-cell pseudotime trajectories were constructed with the Monocle two package (v2.10.1) (*Qiu et al., 2017*) according to the provided documentation (http://cole-trapnell-lab.github.io/monocle-release/). UMI counts were modeled as a negative binomial distribution. The ordering genes were identified as having high dispersion across cells (mean_expression >= 0.01; dispersion_empirical >= 1). The discriminative dimensionality reduction with trees (DDRTree) method was used to reduce data to two dimensions. Differentially expressed genes were identified and used for dynamic trajectory analysis (NO discovery rate [FDR]<0.01) to order cells in pseudotime. The *plot_pseudotime_heatmap* function was used to generate heatmaps.

## NIF analysis

To define NIFs, Refseq-defined transcripts were first converted into Ensemble transcript IDs and their sequences were obtained using the UCSC Table Browser. NIFs were identified in these transcripts using an algorithm written in Python 2.7, Zenith.py, created by the Wilkinson laboratory. Only transcripts with a detectable 5'UTR and 3'UTR were considered. A transcript was defined as harboring a dEJ if it contained at least one exon-exon junction $\geq$50 nt downstream of the stop codon terminating the main ORF. A transcript was defined as harboring an uORF if the following criteria were met: (i) the ORF is in the 5' UTR, (ii) the start codon and surrounding nts are in a context known to initiate translation (a purine at the $-3$ position or a guanine at the $+4$ position, relative to the A in the AUG initiation codon [+1]) (*Kozak, 1986*), (iii) the ORF is $\geq$30 nt long, and (iv) the ORF does not overlap with the main ORF (to reduce the probability that translation could be re-initiated, thereby allowing the transcript to escape NMD).

## Immunofluorescence analysis

Adult mice were anesthetized and perfused with 4% paraformaldehyde (PFA; Sigma). OE was dissected and fixed in 4% PFA at 4°C for 24 hr, then transferred to 70% ethanol. After embedding in paraffin, 5 µM sections were prepared, deparaffinized 2 times in xylene, followed by serial dilutions of ethanol. Unmasking was performed with IHC-TekTM epitope retrieval solution using a steamer (IHCWORLD) for 40 min. Blocking was performed by incubating with 5% serum (from the species that the secondary antibody was raised in) for 1 hr at room temperature. The sections were then

incubated overnight with the primary antibody (goat polyclonal OMP, rabbit anti-CAMP [*Gallo et al., 1997*]) at 4°C and incubated with secondary antibody (Donkey anti-Goat IgG [H+L] conjugated with Alexa Fluor 488 or Donkey anti-Rabbit IgG [H+L] conjugated with Alexa Fluor 555) for 1 hr at room temperature. The nuclei were counterstained with DAPI, and a coverslip was placed over the sections with mounting medium.

## Western blot analysis

OEs were incubated in radioimmunoprecipitation assay (RIPA) buffer (Bio-Rad) supplemented with protease inhibitor cocktail (Sigma) on ice for 30 min, followed by centrifugation at 16,000 *g* for 15 min at 4°C. The lysates were then transferred to new tubes, and protein level was quantified using the DC Protein Assay kit (Bio-Rad). Twenty micrograms of the protein samples were separated on an 15% polyacrylamide gel, and Western blot analysis was performed as previously described (*Ramaiah et al., 2019*). Quantification of the blots was performed using NIH ImageJ (1.8.0).

## Acknowledgements

We thank the UCSD Institute for Genomic Medicine for technical support and the San Diego Supercomputer Center for providing data analysis resources. We are grateful to Drs. Maike Sander (UCSD), Haiqing Zhao (Johns Hopkins University), and Paul Ameiux (University of Washington) for providing R26-eYFP, *Omp-Cre*, and RiboTag mice, respectively.

## Additional information

### Funding

| Funder | Grant reference number | Author |
| --- | --- | --- |
| Eunice Kennedy Shriver National Institute of Child Health and Human Development | R01 HD093846 | Miles F Wilkinson |

The funders had no role in study design, data collection and interpretation, or the decision to submit the work for publication.

### Author contributions

Kun Tan, Resources, Data curation, Formal analysis, Validation, Writing - original draft; Samantha H Jones, Conceptualization, Data curation; Blue B Lake, Jennifer N Chousal, Formal analysis; Eleen Y Shum, Conceptualization; Lingjuan Zhang, Song Chen, Abhishek Sohni, Shivam Pandya, Data curation; Richard L Gallo, Kun Zhang, Heidi Cook-Andersen, Supervision; Miles F Wilkinson, Conceptualization, Supervision, Funding acquisition, Project administration, Writing - review and editing

### Author ORCIDs

Kun Tan https://orcid.org/0000-0002-8567-7795
Blue B Lake http://orcid.org/0000-0002-8637-9044
Song Chen http://orcid.org/0000-0001-5286-3084
Miles F Wilkinson https://orcid.org/0000-0002-6416-3058

### Ethics

Animal experimentation: This study was carried out in strict accordance with the Guidelines of the Institutional Animal Care and Use Committee (IACUC) at the University of California, San Diego. The protocol was approved by the IACUC at the University of California, San Diego (permit number: S09160).

### Decision letter and Author response

Decision letter https://doi.org/10.7554/eLife.57525.sa1
Author response https://doi.org/10.7554/eLife.57525.sa2

# Additional files

## Supplementary files

- Supplementary file 1. Quality control matrices of Upf3b-null and control mOSNs RNA-seq datasets. (**1**) QC metrics, (**2**) reads count, and (**3**) TPM values from *Upf3b*-null (KO) and control (WT) mOSN RNA-seq and RiboTag datasets. (**4**) The expression of all annotated *Olfr* genes in *Upf3b*-null (KO) and control (WT) mOSN samples, based on RNA-seq analysis.

- Supplementary file 2. mOSN transcriptome, translome, and NMD targets. (**1**) Genes differentially expressed between *Upf3b*-null and WT mOSNs (as defined by RNA-seq analysis), including their known NIFs. (**2**) High-confidence NMD targets: transcripts both upregulated and stabilized in *Upf3b*-null mOSNs. (**3**) The transcriptome and translome of *Upf3b*-null and WT mOSNs, based on RNA-seq and RiboTag analyses. (**4**) Translational efficiency (TE) of RNAs expressed in *Upf3b*-null and WT mOSNs, as defined by RNA-seq and RiboTag analyses.

- Supplementary file 3. Putative mouse NMD target RNAs defined by previous studies.

- Supplementary file 4. Genes exhibiting enriched expression in scRNA-seq-defined OE cell subsets. (**1**) Genes exhibiting enriched expression in all OE cell types identified by scRNA-seq analysis. (**2**) Genes exhibiting enriched expression in HBC, GBC, iOSN, and mOSN sub-clusters identfied by scRNA-seq analysis. (**3**) Genes exhibiting enriched expression in the 4 temporally distinct gene groups defined by pseudotime trajectory analysis of HBCs, GBCs, iOSNs, and mOSNs. (**4**) Transcription factor (TF) genes exhibiting enriched expression in the 4 temporally distinct gene groups defined by pseudotime trajectory analysis of HBCs, GBCs, iOSNs, and mOSNs.

- Supplementary file 5. Immune genes expressed in HBCs, GBCs, iOSNs, and mOSNs. (**1**) Immune genes enriched in *Upf3b*-null HBC, GBC, iOSN, and mOSN sub-clusters identified by scRNA-seq analysis. (**2**) NIFs present in immune genes expressed in *Upf3b*-null OSNs.

- Transparent reporting form

## Data availability

Sequencing data have been deposited in GEO under accession code GSE146043.

The following dataset was generated:

| Author(s) | Year | Dataset title | Dataset URL | Database and Identifier |
|---|---|---|---|---|
| Tan K, Jones SH, Wilkinson MF | 2020 | The role of the NMD factor UPF3B in olfactory sensory neurons | https://www.ncbi.nlm.nih.gov/geo/query/acc.cgi?acc=GSE146043 | NCBI Gene Expression Omnibus, GSE146043 |

The following previously published datasets were used:

| Author(s) | Year | Dataset title | Dataset URL | Database and Identifier |
|---|---|---|---|---|
| Hurt JA, Robertson AD, Burge CB | 2013 | Global Analyses of UPF1 Binding and Function Reveal Expanded Scope of Nonsense-Mediated mRNA Decay | https://www.ncbi.nlm.nih.gov/geo/query/acc.cgi?acc=GSE41785 | NCBI Gene Expression Omnibus, GSE41785 |
| Mao H, McMahon JJ, Tsai YH, Wang Z, Silver DL | 2016 | Haploinsufficiency for Core Exon Junction Complex Components Disrupts Embryonic Neurogenesis and Causes p53-Mediated Microcephaly | https://www.ncbi.nlm.nih.gov/geo/query/acc.cgi?acc=GSE85576 | NCBI Gene Expression Omnibus, GSE85576 |
| Mooney CM, Jimenez-Mateos EM, Engel T, Mooney C, Diviney M, Venø MT, Kjems J, Farrell MA, O'Brien DF, Delanty N, Henshall DC | 2017 | RNA sequencing of synaptic and cytoplasmic Upf1-bound transcripts supports contribution of nonsense-mediated decay to epileptogenesis | https://doi.org/10.1038/srep41517 | RNA sequencing of synaptic and cytoplasmic Upf1-bound transcripts supports contribution of nonsense-mediated decay to epileptogenesis, 10. |

| | | | | | |
|---|---|---|---|---|---|
| | | | | | 1038/srep41517 |
| Bao j, Vitting-Seer-up K, Waage J, Tang C, Ge Y, Porse BT, Yan W | 2016 | UPF2-Dependent Nonsense-Mediated mRNA Decay Pathway Is Essential for Spermatogenesis by Selectively Eliminating Longer 3'UTR Transcripts | https://www.ncbi.nlm.nih.gov/geo/query/acc.cgi?acc=GSE55180 | | NCBI Gene Expression Omnibus, GSE55180 |
| McIlwain DR, Pan Q, Reilly PT, Elia AJ, McCracken S, Wakeham AC, Itie-Youten A, Blen-cowe BJ, Mak TW | 2010 | Smg1 is required for embryogenesis and regulates diverse genes via alternative splicing coupled to nonsense-mediated mRNA decay | https://doi.org/10.1073/pnas.1007336107 | | Smg1 is required for embryogenesis and regulates diverse genes via alternative splicing coupled to nonsense-mediated mRNA decay, 10.1073/pnas.1007336107 |
| Weischenfeldt J, Damgaard I, Bryder D, Theilgaard-Mönch K, Thoren LA, Nielsen FC, Jacobsen SEW, Nerlov C, Porse BT | 2008 | NMD Is Essential for Hematopoietic Stem and Progenitor Cells and for Eliminating By-Products of Programmed DNA Rearrangements10.1101/gad.468808 | NMD Is Essential for Hematopoietic Stem and Progenitor Cells and for Eliminating By-Products of Programmed DNA Rearrangements, 10.1101/gad.468808 | | |
| Thoren LA, Nør-gaard GA, Weischenfeldt J, Waage J, Jakobsen JS, Damgaard I, Bergström FC, Blom AM, Borup R, Bisgaard HC, Porse BT | 2010 | UPF2 Is a Critical Regulator of Liver Development, Function and Regeneration | https://doi.org/10.1371/journal.pone.0011650 | | UPF2 Is a Critical Regulator of Liver Development, Function and Regeneration, 10.1371/journal.pone.0011650 |

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
