## [Decision Letter]

Thank you for submitting your article "The NMD factor UPF3B shapes olfactory neurogenesis" for consideration by *eLife*. Your article has been reviewed by three peer reviewers, and the evaluation has been overseen by Didier Stainier as the Senior and Reviewing Editor. The following individuals involved in review of your submission have agreed to reveal their identity: David M Bedwell (Reviewer #1); Luis R Saraiva (Reviewer #2).

The reviewers have discussed the reviews with one another and the Reviewing Editor has drafted this decision to help you prepare a revised submission.

*Reviewer #1:*

In this study, the authors examined the role of the NMD factor UPF3B in olfactory neurogenesis. They identified stage-specific olfactory sensory neuron (OSN) cell clusters in WT mice and found that UPF3B loss resulted in changes in the proportion of these cell clusters. They also identified a number of interesting UPF3B-dependent NMD target transcripts in vivo using single-cell RNA-sequencing (scRNAseq) analysis, including strong candidates with roles in olfactory neurogenesis, and neural development. Finally, they showed that UPF3B dampens the expression of a large cadre of anti-microbial genes and promotes the selection of specific olfactory receptor (*Olfr*) genes. This study provides important new insights into the olfactory field and identifies an important new post-transcriptional regulator that shapes OSN gene expression and OSN development.

Overall, this is an interesting manuscript with exiting new insights into the role of UPF3B in regulating gene expression in OSN cell clusters. However, I suggest that the authors consider the following concerns:

1) I am concerned about the lack of validation of your RNA-seq data in Figure 1. The only secondary validation seemed to be the experiment using the method of Alkallas et al. to measure the stability of 127 mRNAs upregulated in *Upf3b*-null mOSNs. You state that this method "infers RNA stability based on pre-mRNA and steady-state mRNA levels". Going to the Alkallas paper, they say that it is "a method for unbiased estimation of differential mRNA decay rate from RNA-sequencing data by modeling the kinetics of mRNA metabolism." So, it seems to uphold the changes in abundance as predicted by the RNA-seq, but it seems that you are using RNA-seq to validate RNA-Seq data, which seems to be a circular argument. I didn't see any other data examining the validity of your RNA-seq data. Is there not any other way to confirm these results from the FACS-purified mOSNs?

2) The labels of many panels (F-J) in Figure 2 do not match the text or figure legend. This made it difficult to try to match the right panel with the description in the text (and legend). Please correct the labeling.

*Reviewer #2:*

The authors apply a mixed bulk and single-cell RNA-seq based approach to address an interesting and important question regarding the role of NMD in the neurogenesis and development of OSNs in the mouse nose. As it stands, the structure and language used in many sections of this manuscript made it confusing and hard to follow. Critically, the lack of rigor in the statistical and methodological reporting impact my ability to properly assess the validity of the results and conclusions of this study.

Thus, there are several points that I would like to see the authors address:

1) Based the body of olfaction-related literature cited, study design and naïveté of the analysis, it is clear that the authors are not well familiarized with the olfactory system. In the last 6 years, multiple studies have performed RNA-seq in the mouse whole OE, pools of sorted OSNs and also single OSNs and other cell types (PMIDs: 25187969, 31392275, 26670777, 26646940, 26541607, 28506465). I was surprised to note that with the exception of the above underlined reference, the authors failed to mention and discuss this work in their submitted manuscript. Why is this important? Some key results from those studies would have helped guide the experimental design and data analysis presented in this submitted manuscript (see point 2).

2) Some of the key results in this manuscript arise from FACS experiments done using dissociated cells from the nose of OMP-YFP mice, but details are missing regarding the number of cells sorted from each mouse, and also the number of cells (and animals) that incorporate each sample sequenced. Why is this important? Each mouse has on average 10 million mOSNs, each expressing 1 out of ~1100 intact *Olfr* genes. Since *Olfr* genes are expressed in the OE in a zonal fashion, dissection, dissociation and FACS experimental procedures can all lead to subsampling and subsequently severely bias the results. Moreover, it is known from previous FACS+RNA-seq experiments using OMP-GFP animals, that 2 sub-populations exist among the mOSNs expressing *Omp*, one expressing lower *Omp* levels than the other, and representing mOSNs that are not yet in the latest maturation stage possible (PMID: 26670777). Thus, in order to critically assess the validity of the conclusions from the analyses relative to Figure 1 and Figure 2, the authors should provide the following information:

2.1) The exact number of sorted cells for each of their sequenced samples, and FACS plots (and associated gating cutoffs used) in supplementary data. This could also help explain the puzzling result of why half of the KO and WT samples share the same space in the PCA from Figure 1—figure supplement 1B.

2.2) How many *Olfr* genes (list pseudogenes and intact genes) can the authors detect in each of the sequenced samples and how do they compare between all samples? A sample capturing the whole diversity of the OE should show evidence for expression for 98.9% of all annotated *Olfr* genes (PMID: 26670777). If this is not the case, the downstream analysis is compromised and the conclusions not well supported.

3) The Materials and methods section severely lacks rigor and structure. Some methodological sections need more detail and structure. For example, for every experiment described, the age, number of replicates, genotype and sex of the animals used should be stated. Also, explicit details are lacking throughout the manuscript regarding which exact mouse crossings (for the genetically modified lines) were used in different experiments. More importantly, methods describing some experiments shown in the manuscript are entirely missing from the manuscript. For example, how were the OE samples from WT and KO mice collected and processed for whole tissue RNA-seq and for immunofluorescence? how were the weighting experiments from (Figure 1A) performed? How were the behavioral tests in Figure 1—figure supplement 1A performed? How were the samples prepared ahead of FACS and how many cells, and from how many animals, are contained in each biological replicate of FACS-purified OSNs? How were Western blots performed?

4) The authors mention they analyzed several neuronal markers, which they show in Figure 1B. Not all markers tested are for neurons. In reality, they probed the gene expression for 3 canonical markers of iOSNs (*Gap43*), HBCs (*Krt5*) and mOSNs (*Gnal*). Please correct the main text accordingly. Also, since the authors performed RNA-seq on the whole OE, the RNA-seq gene expression estimates for the canonical markers of the main cell types populating the OE should be shown (I suggest plotting something similar to similar to Figure 1C in PMID: 31392275).

4.1) In this context, since *Omp* is the canonical marker for all mOSNs subtypes, I am puzzled as to why the authors used *Gnal* instead. This is especially relevant because they show the RQ values for *Omp* in Figure 1—figure supplement 1C, but no significant difference was observed between WT and KO, which does not fit with the results from the other markers used in Figure 1B. Please explain.

4.2) Also, it is clear from the immunochemistry experiments in Figure 6G that the layer containing OMP+ cells is thicker in WT vs. KO mice, suggesting that a higher number of OMP+ cells exist in WT mice (in line with the hypothesis that KO mice have reduced olfactory abilities, possibly caused to lower numbers of mOSNs). The authors should check if OMP+ cells are more highly prevalent throughout the whole OE in WT vs. KO, or if this decrease is zonally restricted. A counting of OMP+ per mm^2^ would help convince me and clarify this topic.

5) The authors mention: "Despite the reduction in GBCs and iOSNs, *Upf3b*-null mice had normal numbers of mOSNs (53% vs. 47% in the control) (Figure 6A). Despite the reduction in GBCs and iOSNs, *Upf3b*-null mice had normal numbers of mOSNs (53% vs. 47% in the control) (Figure 6A). Consistent with this, the mOSN marker, OMP, was similarly expressed (at both the RNA and protein levels) in OE from *Upf3b*-null and control mice (Figure 1—figure supplement 1C, D)". How can the authors exclude the hypothesis that potential biases caused by dissection and dissociation (see point 2 above), are the cause for these differences? Do the authors have statistics like shown in Figure 6A for individual mice, or were all the WT and KO samples pulled together in a single experiment for each genotype?

6) The authors mention that in Figure 6G "immunofluorescence analysis showed that an anti-CAMP antiserum showed a strong signal in the OE, as well as the lamina propria". While the presence of CAMP staining in the lamina propria is clear, I cannot distinguish the signal in the OE. Please provide evidence of this or rephrase.

7) It is critical the authors should validate (at least) a subset of the key differentially expressed genes (namely some *Olfrs*) identified in their RNA-seq analyses. This is important to confirm the validity of the RNA-seq data and refute the possibility that the differentially expressed genes identified are due to biases in dissection and/or dissociation. This validation could be done with immunohistochemistry or with in-situ hybridization, followed by counting of the positive cells across the whole OE in multiple animals and the two genotypes.

8) In its current form, the manuscript is not easy to follow. In my opinion, this is partly caused by the structure and language used in several sections of this manuscript, and partly because of the way data is presented in the main figures. In the revised version, the authors should consider simplifying the language to make it more accessible and improving the structure and the way the data is presented, as means to improve clarity (I give several examples on this topic below).

*Reviewer #3:*

The manuscript "The NMD factor UPF3B shapes olfactory neurogenesis" by Tan, Jones and Wilkinson studies the role of UPF3B-dependent NMD in olfactory epithelium (OE). This branch of the nonsense-mediated RNA decay (NMD) pathway is critical for human cognition. Mutations in *Upf3b* have been described as a potential cause of mental retardation, autism, childhood-onset schizophrenia, bipolar disorder and attention-deficit hyperactivity disorder in several families. Using *Upf3b*-null mice, the authors perform the powerful single-cell RNAseq analysis in OE and delineate the roles of UPF3B-dependent NMD in various stages of OE neurogenesis. First, by showing that several olfactory neural markers exhibited significantly decreased expression in *Upf3b*-null as compared to control OE, the authors confirm an olfactory defect upon loss of UPF3B protein. Their RNAseq data on sorted mOSNs identifies more than 200 differentially expressed genes between *Upf3b*-null and control mOSNs. Several genes involved in neurogenesis were among these genes suggesting a role of UPF3B in OE neurogenesis. They then screened the statistically upregulated genes for features known to trigger NMD. By combining this screen with a recently developed method that allows unbiased estimation of mRNA decay, the authors determined likely NMD targets within mOSNs. The authors also provide new cellular and molecular information on wild type OSNs and their development in vivo. Although this part of the study is not directly related to NMD, it provides a unique set of resource for the olfactory field. The scRNAseq analysis was employed to identify novel mOSN cell subsets, and thereby determination of the specific cell subsets of mature olfactory sensory neurons that are regulated by UPF3B-dependent NMD. Their experiments are in-depth and their findings are novel. While I have much enthusiasm for this study, there are some issues that need to be attended to prior to consideration for publication:

Specific comments:

1) The authors should consider changing the title, maybe using a more specific term such as “olfactory epithelium neurogenesis”, as olfactory neurogenesis might be confused with the neurogenesis ongoing in olfactory bulb in adulthood (constant neuron production for olfactory bulb from SVZ). Same rewording should be applied to throughout the text as “olfactory neurogenesis” is broadly used in the manuscript.

2) The authors found that 15 previously defined likely mouse NMD target mRNAs (based on upregulation and/or stabilization in response to NMD factor depletion or high UPF1 occupancy) overlapped with mRNAs upregulated in *Upf3b*-null mOSNs: *Atp10d*, *Lbh*, *Slc38a6*, *Tgm2*, *Rgl3*, *Notch2*, *Ywhag*, *Luc7l*, *Ptch1*, *1700025G04Rik*, *Tle3*, *Ptprn*, *Ptger2*, *Dhps*, and *Msrb3*. To assess the direct NMD targets, the authors measured the stability of the 127 mRNAs upregulated in *Upf3b*-null mOSNs using a method that infers RNA stability based on pre-mRNA and steady-state mRNA levels. This method revealed that 82 of 127 upregulated genes encode mRNAs stabilized in *Upf3b*-null mOSNs as compared to control mOSNs. Of these 82 stabilized and upregulated mRNAs, 52 had at least 1 of the 3 NMD-inducing features (NIFs) (1- an exon-exon junction >50 nt downstream of the stop codon, 2- an uORF, or 3- a long 3'UTR) that they examined. Thus, the authors classified these 52 mRNAs as high confidence mOSN NMD targets.

4 of the 15 previously defined NMD targets (*Tle3*, *Ptprn*, *Rgl3*, and *Dhps*) that initially overlapped with the list of 127 upregulated mRNAs, never made to the list of 52 upregulated, stabilized, and with NIFs. The authors should provide an explanation for this discrepancy. Is it because those 4 genes were identified as NMD targets solely based on their upregulation in response to NMD factor depletion in previous studies? If so, that list or at least those 4 genes should be omitted from the text. I can see that the authors were trying to do a thorough analysis by combining stability, upregulation and NMD-inducing features to determine the direct targets of NMD. However, those 15 previously defined likely mouse NMD target mRNAs are confusing.

3) Presence of uORF and an exon-exon junction >50 nt downstream of the stop codon are well-established NIFs. Is a long 3' UTR alone a well-established NIF? Although the authors used a stability analysis, they should be careful with their interpretation of direct NMD targets, especially with the exon-length criteria. The majority of the direct-target list (34 mRNAs of 52 mRNAs) did not have either a uORF or an exon-exon junction >50 nt downstream of the stop codon, but just a long 3'UTR. They should either provide additional validation for these 34 mRNAs or rephrase the relevant part. Intriguingly, 8 of the previously "defined" 15 NMD-target mRNAs mentioned above were among the 34 mRNAs with long 3'UTRs, while only 3 mRNAs with uORF were common in both lists. Were those 8 mRNAs found to have high UPF1 occupancy or stability upon depletion of a different NMD factor than UPF3B? Because this could provide additional support that they could be direct NMD targets.

4) The four subtypes in olfactory epithelium, and their trajectory (HBC → GBC → iOSN → mOSN) are well established. However, the authors spent more than 2 pages in describing OE cell subsets and molecular pathways active during olfactory neurogenesis. Although *Upf3b*-null mice have reduced numbers of HBCs, more GBCs, and more iOSNs, *Upf3b*-null mice still had normal numbers of mOSNs (53% vs. 47% in the control). Because the authors focused only in OSNs when identifying direct targets of NMD (due to the OSNs being the functional unit in OE), the part describing other OE cell subsets and molecular pathways active during olfactory neurogenesis should be more concise.

5) Table S1 has overwhelming data (5 excel sheets), and it is hard to pull out the antimicrobial genes from 12,000 genes. How many are they? Are they more than what is shown in Figure 6E? If so, it should be shown in a table (similar format to Table 1).

6) The authors suggest a model that UPF3B protein promotes the selection of the downregulated 78 *Olfr* genes to be the dominant *Olfr* gene expressed in individual mOSNs. Is there a transcription factor or other kind of transcriptional regulator that positively regulates *Olfr* genes and is downregulated in KO mOSNs? Did any known repressor of *Olfr* gene expression show an increased expression in KO mOSNs?

7) The scRNAseq data shows that the *Upf3b*-null OEs lack mOSN-2 sub-cluster. Could the downregulation of 78 *Olfr* genes in KO mOSNs be a reflection of this? Maybe, those 78 genes are predominantly expressed in the mOSN-2 sub-cluster. The authors should discuss this.

8) The Western blot of CAMP looks convincing. However, it is not specific to the OSNs. The CAMP signal is not clear in mOSNs in immunohistochemistry images. It looks like that CAMP was upregulated in the entire OE, signal being weak in mOSNs compared to other cell types. Showing CAMP staining only with DAPI in another panel (no merging with OMP staining) would work.

9) In addition to *Olfr* genes, the authors discover that UPF3B in OSNs also regulates antimicrobial genes. Unlike the *Olfr*s, the antimicrobial genes are upregulated in KO mOSNs. Similar to comment #6, is any of the direct NMD targets identified in Figure 2 known to trigger inflammation or antimicrobial genes? The authors should offer a mechanism/s as to how NMD indirectly regulates these major sets of genes. Their finding of another major class of genes is regulated by UPF3B suggests that the UPF3B prortein has independent functions within OSNs, one being suppression of OE inflammation. Or, can loss of *Olfr* genes and an increase in immune response be connected? Can one be responsible for the other?

[Editors' note: further revisions were suggested prior to acceptance, as described below.]

Thank you for submitting your article "The role of the NMD factor UPF3B in olfactory sensory neurons" for consideration by *eLife*. Your article has been reviewed by three peer reviewers, and the evaluation has been overseen by Didier Stainier as the Senior and Reviewing Editor. The following individuals involved in review of your submission have agreed to reveal their identity: David M Bedwell (Reviewer #1); Luis R Saraiva (Reviewer #2).

While the revised manuscript was found to be much improved, there are several remaining issues that should be addressed.

*Reviewer #1:*

In this study, the authors examined the role of the NMD factor UPF3B in olfactory neurogenesis. They identified stage-specific olfactory sensory neuron (OSN) cell clusters in WT mice and found that UPF3B loss resulted in changes in the proportion of these cell clusters. They also identified a number of interesting UPF3B-dependent NMD target transcripts in vivo using single-cell RNA-sequencing analysis, including strong candidates with roles in olfactory neurogenesis, and neural development. Finally, they showed that UPF3B dampens the expression of a large cadre of anti-microbial genes and promotes the selection of specific olfactory receptor (*Olfr*) genes. This study provides important new insights into the olfactory field and identifies an important new post-transcriptional regulator that shapes OSN gene expression and OSN development.

Overall, this is an interesting manuscript with exciting new insights into the role of UPF3B in regulating gene expression in OSN cell clusters. The authors adequately addressed each of my previous concerns.

*Reviewer #2:*

The authors have made improvements in the manuscript, but unfortunately, they have not convincingly addressed some of my major concerns. Below I am detailing my responses to the rebuttal letter point by point:

1) No more comments. This concern is addressed.

2.1) The authors then present "two lines of evidence against a significant subsampling bias in their samples: First, each sample was pooled from 3 mice. Second, most samples had a similar percentage of OMP-YFP+ cells (between 5-7%)".

Unfortunately, without knowing the numbers of cells sorted in each sample, I am failing to see how any of these two reasons argue against it. The olfactory epithelium of young adult mice (8-10 weeks) is populated on average by ~10 million mature OSNs (mOSNs). If each sample contains on average 10,000 sorted cells, that would capture only 0.1% of all mOSNs. If each sample contains 100,000 sorted cells, that would still only make up 1% of all mOSNs, and so on. Since the authors "did not make note of the output cell number" in each sample, how could pooling animals or sorting similar percentages of cells between animals be used as an argument against subsampling? Also, only 4/8 of their samples had between 5-7% of sorted OMP-YFP+ cells, which is not "most samples". I am assuming the authors kept the FACS files, and thus should be able to retrieve those values. Is this the case?

Finally, what is the logic underlying the following statement: "We note that our manuscript is currently ~30% over the word limit."?

2.2) The authors here claim that they asked, "what genes are regulated by a particular factor in mOSNs" and that "the depth of RNA-seq analysis we used was sufficient to draw several conclusions". One of the unexpected major conclusions of the authors is that "over half of the downregulated genes in Upf3-null mOSNs are *Olfr* genes" and state that they "regard this finding as a robust result, as the RNA-seq coverage for *Upf3b*-null and control OSNs was 95.8% and 96.0%, respectively, of all annotated *Olfr* gene".

Since *Olfr* and many other genes are expressed zonally in the olfactory epithelium (PMID: 7812149, 15814789, 12709059, 32209480, etc), differences in dissection between animals could easily account for the differences observed in the RNA-seq data from sorted cells. Thus, showing the gene expression distribution of all *Olfr* genes for each of the sorted samples is a key QC step for this study and it should feature early on in the main figures. The authors should include that in the main text and present a plot containing the mean gene expression values (in log10) for all annotated mouse *Olfr* in either descending or ascending order for the WT and KO, include similar plots in supplementary for each individual sample and include all the gene names (and not just Ensembl IDs) in the “TPM” sheet of Supplementary file 1 to make this accessible to the community. It would also be very useful to present in supplementary a Venn diagram comparing all the *Olfr*s expressed in WT vs KO mice.

3) The Materials and methods are now much improved. Only one minor comment here: the authors should include the commercial source of the coyote and bobcat urines.

4) As I mentioned in my original comment, as *Gap43* and *Krt5* are not expressed in mature OSNs, they should not be called “olfactory neuronal markers”. Instead, *Gap43* is a canonical marker for immature neurons and *Krt5* a canonical marker for horizontal basal cells. In line with this, the authors do mention that "mOSN marker gene, *Omp*, is enriched in these mOSN samples, and that the non-mOSN markers genes, *Krt5* and *Lgr5*, are de-enriched in these samples". Finally, the authors mention that their RNA-seq analysis was done "purified mOSNs, not whole OE". I was indeed aware of this, as it is was already explicit in the manuscript. I suggested plotting something similar to Figure 1C in PMID: 31392275, for 2 main reasons: i) it summarizes in one plot the gene expression levels for the canonical markers for all major cell types populating the olfactory epithelium, and ii) it visually allows the reader assess the levels of “contamination/hitchhiking” by other cell types during the sorting procedure.

Please correct this by specifically stating the cell types that those genes are indicative of (also, by “de-enriched” I assume the authors mean “depleted”?).

4.1) The *Omp* expression data should feature in Figure 1B, as it is the best canonical marker for mOSNs (also see previous response). It is well established that *Omp* and *Gnal* are expressed ubiquitously in the majority of mOSNs across all the olfactory epithelium, so I do not see how that could be an explanation.

4.2) No more comments. This concern is addressed.

5) Regarding Figure 6—figure supplement 1B in the revised manuscript: HBCs and GBCs are not OSNs, and it is incorrect to refer them as such. Only immature and mature OSNs should be called OSNs. Please correct this in the figures and throughout the text. Also, on the right-hand panel of Figure 6—figure supplement 1B it would be useful to list all the other cell types that make up 100%.

6) No more comments. This concern is addressed.

7) *Olfr* genes are expressed zonally in the olfactory epithelium (PMID: 7812149, 15814789, 12709059, 32209480, etc) and differences in dissection between animals could easily translate into differential expression artifacts. Moreover, different mouse strains express different *Olfrs* at different levels (PMID: 28438259). Since the authors i) have not yet convincingly addressed my previous concerns about a potential bias in dissection (see above), ii) the mice used in this studies were on mixed or different genetic backgrounds, and iii) one of the major claims of the authors is that half of the differentially expressed genes are *Olfrs*; the authors should validate at least their top 2 differentially expressed *Olfrs* across the two genotypes.

8) This comment is addressed.

*Reviewer #3:*

The authors have done an excellent job revising this manuscript. The revised submission is significantly improved. I have no further comments. The manuscript should be accepted as it is.

---

## [Author Response]

Reviewer #1:[…]1) I am concerned about the lack of validation of your RNA-seq data in Figure 1. The only secondary validation seemed to be the experiment using the method of Alkallas et al. to measure the stability of 127 mRNAs upregulated in Upf3b-null mOSNs. You state that this method "infers RNA stability based on pre-mRNA and steady-state mRNA levels". Going to the Alkallas paper, they say that it is "a method for unbiased estimation of differential mRNA decay rate from RNA-sequencing data by modeling the kinetics of mRNA metabolism." So, it seems to uphold the changes in abundance as predicted by the RNA-seq, but it seems that you are using RNA-seq to validate RNA-Seq data, which seems to be a circular argument. I didn't see any other data examining the validity of your RNA-seq data. Is there not any other way to confirm these results from the FACS-purified mOSNs?

To validate, we performed qPCR analysis on 8 genes with known functions, such as in neural development, from the list of genes shown by our RNA-seq analysis to be differentially expressed between *Upf3b*-null mOSNs and control mOSNs. This qPCR analysis confirmed the regulation of 7 out of 8 of these genes (Figure 1—figure supplement 2B in the revised manuscript). The one exception – *Ptger2* – was not detectably expressed using the qPCR primers we used (data not shown). Genes shown to be upregulated in Upf3-null mOSNs by RNA-seq analysis were found to also be upregulated in *Upf3b*-null mOSNs as judged by qPCR; the same concordance between the 2 assays was shown for downregulated genes.

2) The labels of many panels (F-J) in Figure 2 do not match the text or figure legend. This made it difficult to try to match the right panel with the description in the text (and legend). Please correct the labeling.

We thank reviewer 1 for noticing this. We corrected the labeling in the revised manuscript.

Reviewer #2:[…]Thus, there are several points that I would like to see the authors address:1) Based the body of olfaction-related literature cited, study design and naïveté of the analysis, it is clear that the authors are not well familiarized with the olfactory system. In the last 6 years, multiple studies have performed RNA-seq in the mouse whole OE, pools of sorted OSNs and also single OSNs and other cell types (PMIDs: 25187969, 31392275, 26670777, 26646940, 26541607, 28506465). I was surprised to note that with the exception of the above underlined reference, the authors failed to mention and discuss this work in their submitted manuscript. Why is this important? Some key results from those studies would have helped guide the experimental design and data analysis presented in this submitted manuscript (see point 2).

We thank reviewer 2 for providing the PubMed numbers for these genome-wide studies on the topic of OE cells. We had previously read most of these papers, but not all of them. In our revised manuscript, we cite all 6 of these papers in the Introduction as follows: “To gain insight into the nature of the cells in the OE and their developmental relationships, recent studies have performed transcriptome profiling using whole OE, pools of sorted OSNs, single OSNs, or single OE cells (Fletcher et al., 2017; Ibarra-Soria, Levitin, Saraiva, and Logan, 2014; Saraiva et al., 2015; Saraiva et al., 2019; Tan, Li, and Xie, 2015). […] These studies have also advanced our understanding of the evolution of mammalian olfaction”.

2) Some of the key results in this manuscript arise from FACS experiments done using dissociated cells from the nose of OMP-YFP mice, but details are missing regarding the number of cells sorted from each mouse, and also the number of cells (and animals) that incorporate each sample sequenced. Why is this important? Each mouse has on average 10 million mOSNs, each expressing 1 out of ~1100 intact Olfr genes. Since Olfr genes are expressed in the OE in a zonal fashion, dissection, dissociation and FACS experimental procedures can all lead to subsampling and subsequently severely bias the results. Moreover, it is known from previous FACS+RNA-seq experiments using OMP-GFP animals, that 2 sub-populations exist among the mOSNs expressing Omp, one expressing lower Omp levels than the other, and representing mOSNs that are not yet in the latest maturation stage possible (PMID: 26670777). Thus, in order to critically assess the validity of the conclusions from the analyses relative to Figure 1 and Figure 2, the authors should provide the following information:

We thank reviewer 2 for bringing up this issue. While we cannot completely rule out a subtle subsampling bias, we provide evidence below in our response to point 2.1, against this possibility.

2.1) The exact number of sorted cells for each of their sequenced samples, and FACS plots (and associated gating cutoffs used) in supplementary data. This could also help explain the puzzling result of why half of the KO and WT samples share the same space in the PCA from Figure 1—figure supplement 1B.

The FACS plots of all the dissociated OE samples and associated gating cut-offs used for our RNA-seq analysis are in Figure 1—figure supplement 1B. Unfortunately, we did not make note of the output cell number. However, two lines of evidence against a significant subsampling bias are the following: First, each sample was pooled from 3 mice. Second, most samples had a similar percentage of OMP-YFP+ cells (between 5-7%). With regard to reviewer 2’s comment above about OMP-high and -low cell populations, we did not detect these two populations, perhaps because we used a different reporter mouse than reviewer 2 did in their analysis (Saraiva et al., 2015). We have not indicated this in the revised manuscript, but can do so if deemed important. We note that our manuscript is currently ~30% over the word limit.

2.2) How many Olfr genes (list pseudogenes and intact genes) can the authors detect in each of the sequenced samples and how do they compare between all samples? A sample capturing the whole diversity of the OE should show evidence for expression for 98.9% of all annotated Olfr genes (PMID: 26670777). If this is not the case, the downstream analysis is compromised and the conclusions not well supported.

The reference brought up by reviewer 2 required “deep analysis” to answer a key question it addresses – what is the nature of the expressed *Olfr* repertoire? This is a fascinating question, as mice harbor >1000 *Olfr* genes and thus it is important to know whether all of these genes are actually expressed. Using RNA-seq analysis, the paper shows that at least 98.9% of all *Olfr* genes in mice are detectably expressed in the OE. Thus, the vast majority of the >1000 *Olfr* genes in the mouse genome are expressed and, thus, have the potential to be functional. To draw this conclusion, it was critical that the authors do extremely deep RNA-seq analysis on a very large number of mOSNs.

In our case, we were instead asking what genes are regulated by a particular factor in mOSNs. The depth of RNA-seq analysis we used was sufficient to draw several conclusions. For example, we were able to define numerous in vivo NMD RNA targets from our RNA-seq analysis. Unexpectedly, our RNA-seq analysis revealed that over half of the downregulated genes in Upf3-null mOSNs are *Olfr* genes (Figure 1C in the revised manuscript). We regard this finding as a robust result, as the RNA-seq coverage for *Upf3b*-null and control OSNs was 95.8% and 96.0%, respectively, of all annotated *Olfr* genes. Given that this coverage is only slightly less that of the study described above (98.9%), it is unlikely that we missed many *Olfr* genes significantly misregulated in *Upf3b*-null OSNs. But even if this case (a caveat we can bring up in our manuscript if deemed important), our data still strongly supports the concept that a surprisingly large fraction of *Olfrs* are regulated by NMD in mOSNs. Furthermore, our data documents 78 specific *Olfrs* subject to this regulation.

3) The Materials and methods section severely lacks rigor and structure. Some methodological sections need more detail and structure. For example, for every experiment described, the age, number of replicates, genotype and sex of the animals used should be stated. Also, explicit details are lacking throughout the manuscript regarding which exact mouse crossings (for the genetically modified lines) were used in different experiments. More importantly, methods describing some experiments shown in the manuscript are entirely missing from the manuscript. For example, how were the OE samples from WT and KO mice collected and processed for whole tissue RNA-seq and for immunofluorescence? how were the weighting experiments from (Figure 1A) performed? How were the behavioral tests in Figure 1—figure supplement 1A performed? How were the samples prepared ahead of FACS and how many cells, and from how many animals, are contained in each biological replicate of FACS-purified OSNs? How were Western blots performed?

We apologize for this previously missing information, which has been added to the revised Materials and methods section.

4) The authors mention they analyzed several neuronal markers, which they show in Figure 1B. Not all markers tested are for neurons. In reality, they probed the gene expression for 3 canonical markers of iOSNs (Gap43), HBCs (Krt5) and mOSNs (Gnal). Please correct the main text accordingly. Also, since the authors performed RNA-seq on the whole OE, the RNA-seq gene expression estimates for the canonical markers of the main cell types populating the OE should be shown (I suggest plotting something similar to similar to Figure 1C in PMID: 31392275).

With regard to the qPCR analysis of whole OE, we corrected the text to say: “we found that olfactory neural markers (*Gap43*, *Krt5*, and *Gnal*) exhibited significantly decreased expression in *Upf3b*-null as compared to control OE (Figure 1B)”. With regard to the RNA-seq analysis, this was done on purified mOSNs, not whole OE – we have revised the manuscript to make this clearer. We showed in Figure 2A in the revised manuscript that the expression of the mOSN marker gene, *Omp*, is enriched in these mOSN samples, and that the non-mOSN markers genes, *Krt5* and *Lgr5*, are de-enriched in these samples.

4.1) In this context, since Omp is the canonical marker for all mOSNs subtypes, I am puzzled as to why the authors used Gnal instead. This is especially relevant because they show the RQ values for Omp in Figure 1—figure supplement 1C, but no significant difference was observed between WT and KO, which does not fit with the results from the other markers used in Figure 1B. Please explain.

In the originally submitted manuscript, we showed *Omp* mRNA and OMP protein expression in the same figure (Figure 6—figure supplement 1C, D in the revised manuscript) to convey – in one place – the evidence that OMP expression was not significantly changed in *Upf3b*-null mice. If deemed important, we could move this data to a main figure. A likely explanation for why *Omp* and *Gnal* respond differently in *Upf3b*-null OE is these mOSN markers label different mOSN sub-populations.

4.2) Also, it is clear from the immunochemistry experiments in Figure 6G that the layer containing OMP+ cells is thicker in WT vs. KO mice, suggesting that a higher number of OMP+ cells exist in WT mice (in line with the hypothesis that KO mice have reduced olfactory abilities, possibly caused to lower numbers of mOSNs). The authors should check if OMP+ cells are more highly prevalent throughout the whole OE in WT vs. KO, or if this decrease is zonally restricted. A counting of OMP+ per mm^2^ would help convince me and clarify this topic.

We thank reviewer 2 for noticing this. The image in Figure 6G in our originally submitted manuscript is not representative with respect to the typical width of the mOSN layer we observed in *Upf3b*-null OE. We provide representative images from three individual mice from each genotype in Figure 6—figure supplement 2B in the revised manuscript. Quantification showed that OMP+ cell density is not significantly between the two genotypes.

5) The authors mention: "Despite the reduction in GBCs and iOSNs, Upf3b-null mice had normal numbers of mOSNs (53% vs. 47% in the control) (Figure 6A). Despite the reduction in GBCs and iOSNs, Upf3b-null mice had normal numbers of mOSNs (53% vs. 47% in the control) (Figure 6A). Consistent with this, the mOSN marker, OMP, was similarly expressed (at both the RNA and protein levels) in OE from Upf3b-null and control mice (Figure 1—figure supplement 1C, D)". How can the authors exclude the hypothesis that potential biases caused by dissection and dissociation (see point 2 above), are the cause for these differences? Do the authors have statistics like shown in Figure 6A for individual mice, or were all the WT and KO samples pulled together in a single experiment for each genotype?

Figure 6—figure supplement 1B shows the fraction of HBCs, GBCs, iOSNs, and mOSN in the individual samples we analyzed by scRNA-seq analysis. With regard to mOSNs, we found their representation differs to some extent between each sample and there is neither an obvious nor a statistical difference (p = 0.91) between the 4 *Upf3b*-null and 4 control samples when compared to all OSNs (Figure 6—figure supplement 1A). The same was the case when we determined the fraction of mOSNs per all OE cells (p = 0.44, Figure 6—figure supplement 1B). These results are in agreement with our finding that there was no statistical difference in OMP protein expression as determined by Western, which was analyzed in OE from different mice than those used for scRNA-seq, thereby increasing confidence in the result (Figure 6—figure supplement 1D in our revised manuscript). That said, we cannot rule out that small differences in the dissection and/or cell dissociation used to generate the samples obscured a subtle change in the fraction of mOSNs in the 2 genotypes. Given that reviewer 2 has considerable experience with scRNA-seq analysis of the OE, they can perhaps appreciate the challenges in this experiment, including that cell isolation must be done extremely quickly, thereby precluding pooling OE from several mice to reduce heterogeneity (by contrast, we pooled OE from 3 mice per sample for RNA-seq analysis). In the revised manuscript, we now explicitly state this as a caveat in our revised manuscript: “However, we cannot rule out that the variability among the 4 samples for each genotype might have obscured a subtle change in the fraction of these other OSN stages in *Upf3b*-null mice. This variability might either be the result of biological differences between individual mice or differences in dissection and/or cell dissociation”. We also did not observe a statistically significant difference in either GBCs or iOSNs: analysis of the individual samples showed no statistical difference between *Upf3b*-null and control mice, when compared to all OSNs (p = 0.49 and 0.90,respectively, Figure 6—figure supplement 1A) or compared to all OE cells (p = 0.59 and 0.94, respectively, Figure 6—figure supplement 1B). We previously suggested an apparent shift in the frequency of both GBCs and iOSNs in the KO, based on a striking shift in their average frequency, but this did not hold up to statistical scrutiny.

With regard to HBCs, statistical analysis showed a significant decrease in their frequency in *Upf3b*-null mice, either when compared to all OSNs (p = 0.04, Figure 6—figure supplement 1A) or compared to all OE cells (p = 0.01, Figure 6—figure supplement 1B). To test the validity of this, we performed IHC staining with the HBC marker, TRP63, and found that the density of TRP63+ cells was significantly less in *Upf3b*-null OE than WT OE (Figure 6—figure supplement 1A ). We have revised the manuscript in light of these new results and analyses.

6) The authors mention that in Figure 6G "immunofluorescence analysis showed that an anti-CAMP antiserum showed a strong signal in the OE, as well as the lamina propria". While the presence of CAMP staining in the lamina propria is clear, I cannot distinguish the signal in the OE. Please provide evidence of this or rephrase.

We agree that the anti-CAMP staining is modest in mOSNs. In the revised manuscript, we edited the description of this data as follows: “As further evidence, immunofluorescence analysis detected modest staining in cells in the OE, as well as strong staining in the in the lamina propria, both of which were increased in *Upf3b*-null mice (Figure 6G and Figure 6—figure supplement 2B)”.

7) It is critical the authors should validate (at least) a subset of the key differentially expressed genes (namely some Olfrs) identified in their RNA-seq analyses. This is important to confirm the validity of the RNA-seq data and refute the possibility that the differentially expressed genes identified are due to biases in dissection and/or dissociation. This validation could be done with immunohistochemistry or with in-situ hybridization, followed by counting of the positive cells across the whole OE in multiple animals and the two genotypes.

We validated our RNA-seq analysis by two independent approaches. We validated using qPCR analysis, as described in answer to reviewer 1, critique 1 (Figure 1—figure supplement 2). For validation at the protein level, we elected to test FUT10, as commercial antibodies were available against this protein and it is an interesting factor that is known to promote maintenance of neural stem cells in an undifferentiated state (Kumar et al., 2013). We performed immunofluorescence analysis on FUT10 expression in OE from 3 mice from both genotypes. We co-stained with the mOSN marker, OMP, to label mOSNs. The results showed that the anti-FUT10 antibody signal is broadly decreased in *Upf3b*-null OE, including in *Upf3b*-null mOSNs (Figure 1—figure supplement 2C). This result is consistent with decreased Fut10 RNA expression in *Upf3b*-null mOSNs, as determined by RNA-seq (log2FC=-0.87).

8) In its current form, the manuscript is not easy to follow. In my opinion, this is partly caused by the structure and language used in several sections of this manuscript, and partly because of the way data is presented in the main figures. In the revised version, the authors should consider simplifying the language to make it more accessible and improving the structure and the way the data is presented, as means to improve clarity (I give several examples on this topic below).

We thank reviewer 2 for bringing up this concern, which we have addressed by making a considerable effort to revise wording throughout the entire manuscript (note: alterations that do not impact science content are not marked with the editing program). As part of this, we revised the introductory sentences for some sections to make the new topic more clear, and when necessary, to transition from the previous topic. Finally, we simplified the description of the OE cell clusters and associated molecular pathways identified by our scRNA-seq analysis, as suggested by reviewer 3.

Reviewer #3:[…]Specific comments:1) The authors should consider changing the title, maybe using a more specific term such as “olfactory epithelium neurogenesis”, as olfactory neurogenesis might be confused with the neurogenesis ongoing in olfactory bulb in adulthood (constant neuron production for olfactory bulb from SVZ). Same rewording should be applied to throughout the text as “olfactory neurogenesis” is broadly used in the manuscript.

We elected to broaden the title of our manuscript to: “The role of the NMD factor UPF3B in olfactory sensory neurons.” In addition, we no longer use the term “olfactory neurogenesis” in the revised manuscript.

2) The authors found that 15 previously defined likely mouse NMD target mRNAs (based on upregulation and/or stabilization in response to NMD factor depletion or high UPF1 occupancy) overlapped with mRNAs upregulated in Upf3b-null mOSNs: Atp10d, Lbh, Slc38a6, Tgm2, Rgl3, Notch2, Ywhag, Luc7l, Ptch1, 1700025G04Rik, Tle3, Ptprn, Ptger2, Dhps, and Msrb3. To assess the direct NMD targets, the authors measured the stability of the 127 mRNAs upregulated in Upf3b-null mOSNs using a method that infers RNA stability based on pre-mRNA and steady-state mRNA levels. This method revealed that 82 of 127 upregulated genes encode mRNAs stabilized in Upf3b-null mOSNs as compared to control mOSNs. Of these 82 stabilized and upregulated mRNAs, 52 had at least 1 of the 3 NMD-inducing features (NIFs) (1- an exon-exon junction >50 nt downstream of the stop codon, 2- an uORF, or 3- a long 3'UTR) that they examined. Thus, the authors classified these 52 mRNAs as high confidence mOSN NMD targets.4 of the 15 previously defined NMD targets (Tle3, Ptprn, Rgl3, and Dhps) that initially overlapped with the list of 127 upregulated mRNAs, never made to the list of 52 upregulated, stabilized, and with NIFs. The authors should provide an explanation for this discrepancy. Is it because those 4 genes were identified as NMD targets solely based on their upregulation in response to NMD factor depletion in previous studies? If so, that list or at least those 4 genes should be omitted from the text. I can see that the authors were trying to do a thorough analysis by combining stability, upregulation and NMD-inducing features to determine the direct targets of NMD. However, those 15 previously defined likely mouse NMD target mRNAs are confusing.

There are many possible explanations for why 4 previously defined putative NMD target mRNAs did not make our list of high-confidence NMD targets in mOSNs. The most obvious possibility is that some or all of these 4 RNAs are not actually NMD targets. In support, none of these 4 RNAs have well-established NMD-inducing features (NIFs) established by the field. The *Ptprn* and *Rgl3* mRNAs we found were upregulated have 3’ UTRs of only 522 and 364 nt, respectively; neither has a short upstream open reading frame (ORF); and neither has an exon-exon junction downstream of the main ORF (based on their ENSEMBL transcript IDs of ENSMUST00000027404 and ENSMUST00000045726, respectively). The other 2 mRNAs we found were upregulated are non-coding isoforms of *Tle3* and *Dhps* (ENSMUST00000159140 and ENSMUST00000129826, respectively) and thus, by definition, lack these 3 NIFs. Furthermore, 3 of these 4 RNAs was defined as a putative NMD target mRNA based only on being upregulated in NMD-deficient mice (Author response table 1). Thus, these 3 may be indirectly regulated by NMD. The one exception, *Tle3*, was defined as a likely NMD target based on (i) upregulation in mouse embryonic stem cells (mESCs) depleted of the NMD factor UPF1, (ii) upregulation in mESCs treated with a protein synthesis inhibitor to block NMD, and (iii) high occupancy of the NMD factor UPF1, based on CLIPseq experiments (Hurt, Robertson, and Burge, 2013). Thus, Tle3 mRNA may be a bona fide NMD target in mESCs. One explanation for why we did not find that Tle3 mRNA is significantly stabilized in *Upf3b*-null mOSNs is that it is targeted by NMD in a cell type- or NMD-factor-specific manner. In support of this, we previously showed that some RNAs downregulated by NMD in some tissues are not downregulated by NMD in other tissues (Huang et al., 2011). In addition, we found that NMD targeting also depends on the NMD branch involved, based on knockdown or knockout of different NMD factors (Huang et al., 2011).

To address the above issue, we now report overlap analysis of our high-confidence NMD target mRNAs (rather than all *Upf3b*-null-upregulated mRNA) with previously identified candidate NMD targets: “We found that 11 of these previously defined likely mouse NMD target mRNAs overlapped with the 52 high-confidence targets identified in our study: *Atp10d*, *Lbh*, *Slc38a6*, *Tgm2*, *Notch2*, *Ywhag*, *Luc7l*, *Ptch1*, *1700025G04Rik*, *Ptger2*, and *Msrb3*.”.

Author response table 1. Previously identified putative NMD target mRNAs not in our mOSN high-confidence NMD target mRNA list.

3) Presence of uORF and an exon-exon junction >50 nt downstream of the stop codon are well-established NIFs. Is a long 3' UTR alone a well-established NIF? Although the authors used a stability analysis, they should be careful with their interpretation of direct NMD targets, especially with the exon-length criteria. The majority of the direct-target list (34 mRNAs of 52 mRNAs) did not have either a uORF or an exon-exon junction >50 nt downstream of the stop codon, but just a long 3'UTR. They should either provide additional validation for these 34 mRNAs or rephrase the relevant part. Intriguingly, 8 of the previously "defined" 15 NMD-target mRNAs mentioned above were among the 34 mRNAs with long 3'UTRs, while only 3 mRNAs with uORF were common in both lists. Were those 8 mRNAs found to have high UPF1 occupancy or stability upon depletion of a different NMD factor than UPF3B? Because this could provide additional support that they could be direct NMD targets.

There is extensive literature demonstrating that a long 3’UTR is sufficient to elicit NMD, including the following papers (Ge, Quek, Beemon, and Hogg, 2016; Hogg and Goff, 2010; Kishor, Ge, and Hogg, 2019). Therefore, we consider the 34 genes encoding mRNAs with long 3’UTR that are upregulated in *Upf3b*-null mOSNs as strong candidates to be NMD targets. As reviewer 3 mentioned, a large number of the NMD targets we identified in mOSNs have long 3’UTRs, which raises the possibility that mOSNs have a predilection for degrading mRNAs with this particular NIF, as we now mention in the manuscript. We also note that “By analogy, evidence suggests that male germ cells preferentially target mRNAs harboring long 3’UTRs for destruction by NMD (Bao et al., 2016).”

As reviewer 3 noted, 8 of the 15 genes overlapping between our high-confidence mouse mOSN NMD target list and the list we established of previously defined putative mouse NMD targets encode mRNAs with long 3’UTRs. As shown in Author response table 2, all of these 8 mRNA were defined on the basis of being upregulated in NMD-deficient tissues or cell lines. In most cases, the core NMD factor gene, *Upf2*, was conditionally ablated in the tissue or cell type under study. In two cases, the NMD gene, Smg1, was mutated. Thus, all were defined on the basis of inactivation of a NMD factor gene different than the one we studied – *Upf3b* – thereby increasing confidence that these are bona fide NMD target mRNAs. We have not brought up the above discussion of these 8 genes in the revised manuscript, but we can add this if deemed important. Our revised manuscript is currently ~30% over the length limit.

Author response table 2. High-confidence NMD targets expressed in mOSNs harboring long 3’UTRs that overlap with putative NMD target mRNAs identified in other cell types by previous studies.

4) The four subtypes in olfactory epithelium, and their trajectory (HBC → GBC → iOSN → mOSN) are well established. However, the authors spent more than 2 pages in describing OE cell subsets and molecular pathways active during olfactory neurogenesis. Although Upf3b-null mice have reduced numbers of HBCs, more GBCs, and more iOSNs, Upf3b-null mice still had normal numbers of mOSNs (53% vs. 47% in the control). Because the authors focused only in OSNs when identifying direct targets of NMD (due to the OSNs being the functional unit in OE), the part describing other OE cell subsets and molecular pathways active during olfactory neurogenesis should be more concise.

We agree we were too verbose and thus we have substantially reduced the length of this section.

5) Table S1 has overwhelming data (5 excel sheets), and it is hard to pull out the antimicrobial genes from 12,000 genes. How many are they? Are they more than what is shown in Figure 6E? If so, it should be shown in a table (similar format to table 1).

We thank reviewer 3 for this suggestion. We separated original Table S1 into three supplementary tables, with the RNA-seq and translome data as one supplementary table, scRNA-seq data as another table, and regulated immune genes as another table. We also added a table (Supplementary file 5 in the revised manuscript) listing all enriched immune-associated genes in different *Upf3b*-null OSN sub-clusters.

6) The authors suggest a model that UPF3B protein promotes the selection of the downregulated 78 Olfr genes to be the dominant Olfr gene expressed in individual mOSNs. Is there a transcription factor or other kind of transcriptional regulator that positively regulates Olfr genes and is downregulated in KO mOSNs? Did any known repressor of Olfr gene expression show an increased expression in KO mOSNs?

To determine whether there are either direct or indirect NMD target mRNAs encoding transcriptional regulators that might act in such a circuit, we screened genes exhibiting significantly altered expression in *Upf3b*-null OSNs for those that encode factors known to regulate *Olfr* gene expression or have binding sites in *Olfr* promoters (Clowney et al., 2011; Dalton, Lyons, and Lomvardas, 2013; Hirota and Mombaerts, 2004; Markenscoff-Papadimitriou et al., 2014; McIntyre, Bose, Stromberg, and McClintock, 2008; Michaloski, Galante, and Malnic, 2006; Wang, Tsai, and Reed, 1997). While our screen did not identify any downregulated genes that fulfilled the above criteria, our screen did identify 2 upregulated genes – *Mafg* and *Irf8* – that fulfilled our criteria. Because these 2 genes are upregulated in *Upf3b*-null mOSNs, they may be direct NMD target mRNAs. Intriguingly, *Mafg* and *Irf8* both encode transcriptional repressors (Igarashi et al., 1994; Salem et al., 2014) that bind O/E consensus sites found in most *Olfr* gene promoters (Michaloski et al., 2006). Thus, *Mafg* and *Irf8* are candidates to act directly downstream of NMD in a regulatory circuit that suppresses the transcription of these 78 *Olfr* genes. *Mafg* is a member of the Maf subfamily of basic leucine-zipper transcription factor genes that encode small proteins containing a b-zip DNA-binding domain, but lack a transactivation domain, and thus members of this family dimerize to form transcriptional repressors (Igarashi et al., 1994). MAFG is best known for its ability to regulate globin transcription in erythroid cells; our results raise the possibility that MAFG also functions in OSNs to regulates *Olfr* genes. IRF8 regulates the development hematopoietic cells; its expression in OSNs raises the possibility that this transcription factor also functions in OSNs.

Together, these findings support a model in which (i) IRF8 and MAFG normally subtly repress the transcription of a subset of *Olfr* genes in OSNs to fine tune their expression and (ii) IRF8 and MAFG are encoded by NMD target mRNAs, so when NMD is disrupted, these transcriptional repressors are overexpressed, leading to reduced expression of their *Olfr* gene targets in developing OSNs. Thus, NMD deficiency would be expected to reduce the probability that these particular ^genes will be chosen to be the “dominant *Olfr* gene” in individual mOSNs, which is precisely what we observed in *Upf3b*-null mice. In the revised manuscript, we have posited this model and provided the supporting evidence (a brief version of what is described above) in the Discussion.

We note it is also possible that *Upf3b* dictates the selection of *Olfr* genes by influencing olfactory epithelial neurogenesis. In support, several genes we found to be UPF3B-regulated have been reported to play essential roles in neurogenesis, including Lrp2, Hk2, *Notch2*, Gdf11, Fos, Ptch1, Spry2, and Cwc22, which we mention in the revised manuscript. Finally, it is possible that *Upf3b* differentially influences the survival of OSNs bearing different OLFRs. Consistent with this, we found the Fos is up-regulated in *Upf3b*-null mOSNs. It has been reported that the induction of Fos in OSNs cause cell apoptosis (Michel, Moyse, Brun, and Jourdan, 1994).

7) The scRNAseq data shows that the Upf3b-null OEs lack mOSN-2 sub-cluster. Could the downregulation of 78 Olfr genes in KO mOSNs be a reflection of this? Maybe, those 78 genes are predominantly expressed in the mOSN-2 sub-cluster. The authors should discuss this.

This is an interesting hypothesis. It predicts that some of these 78 *Olfr* genes will be amongst the genes significantly differentially expressed in the mOSN-2 sub-cluster as compared to the other mOSN sub-clusters. However, refuting this hypothesis, no *Olfr* genes are in this list, as shown in Supplementary file 4 in the revised manuscript. Instead genes enriched in the mOSN-2 sub-cluster include Pten, App, Cnga2, Nrp2, Ncam1, Adcy3, *Gnal*, Atf5, and Gfy, all of which are known to be essential for olfactory epithelium development and/or olfaction.

8) The Western blot of CAMP looks convincing. However, it is not specific to the OSNs. The CAMP signal is not clear in mOSNs in immunohistochemistry images. It looks like that CAMP was upregulated in the entire OE, signal being weak in mOSNs compared to other cell types. Showing CAMP staining only with DAPI in another panel (no merging with OMP staining) would work.

We agree that the anti-CAMP staining in mOSNs is modest. In the revised manuscript, we edited our description of this data to say: “As further evidence, immunofluorescence analysis detected modest staining in cells in the OE, as well as strong staining in the in the lamina propria, both of which were increased in *Upf3b*-null mice (Figure 6G and Figure 6—figure supplement 2B).”. Per reviewer 2 request, we have also shown the same image with only the CAMP and DAPI signal (no OMP) (Author response image 1).

**Author response image 1. sa2fig1:** IF analysis of adult mouse OE sections co-stained with antisera against CAMP (red) and OMP (green). Nuclei were stained with DAPI (blue).

9) In addition to Olfr genes, the authors discover that UPF3B in OSNs also regulates antimicrobial genes. Unlike the Olfrs, the antimicrobial genes are upregulated in KO mOSNs. Similar to comment #6, is any of the direct NMD targets identified in Figure 2 known to trigger inflammation or antimicrobial genes? The authors should offer a mechanism/s as to how NMD indirectly regulates these major sets of genes. Their finding of another major class of genes is regulated by UPF3B suggests that the UPF3B prortein has independent functions within OSNs, one being suppression of OE inflammation. Or, can loss of Olfr genes and an increase in immune response be connected? Can one be responsible for the other?

We thank reviewer 3 for bringing this up, as we neglected to point out that a remarkably large number of the upregulated mRNAs encoding anti-microbial and other immune proteins have the potential to be direct NMD targets. We found that 48 out of 88 mRNAs encoding immune-related proteins that are upregulated in *Upf3b*-null OSNs harbor at least one NIF (Supplementary file 5 in the revised manuscript). Thus, these 48 mRNAs are good candidates to be direct targets of the NMD pathway, as we point out in the revised manuscript.

With regard to indirect regulation, the most promising candidate we found to mediate such regulation is *NOTCH2*, which we identified as being encoded by a high-confidence NMD target mRNA (Figure 2 in the revised manuscript). *NOTCH2* has been shown to regulate the expression of many genes encoding inflammatory mediators and antimicrobial proteins, as reviewed in Shang et al. (Shang, Smith, and Hu, 2016). In addition, two other high-confidence NMD targets – *Bhlhe40* and *Rac2* – encode immune system regulators. These candidate regulatory circuit are discussed in the revised manuscript.

Reviewer 3 also asks whether the increased expression of anti-microbial genes in *Upf3b*-null OSNs might somehow be causally connected with the reduced numbers of mOSNs expressing specific OLFRs in *Upf3b*-null mice. One possibility is that some of the 78 OLFRs less represented in *Upf3b*-null mice bind to odorants that normally suppress immune activation. It is also possible is that these 78 OLFRs are underrepresented in *Upf3b*-null OSNs because loss of *Upf3b* increases the likelihood of apoptosis of these cells, which, in turn induces anti-microbial genes in surviving OSNs. We have not mentioned either of these scenarios in the revised manuscript, as we do not consider them very likely. However, if deemed important, we would be happy to discuss these or other models in a subsequent revision.

[Editors' note: further revisions were suggested prior to acceptance, as described below.]

Reviewer #2:[…]1) No more comments. This concern is addressed.2.1) The authors then present "two lines of evidence against a significant subsampling bias in their samples: First, each sample was pooled from 3 mice. Second, most samples had a similar percentage of OMP-YFP+ cells (between 5-7%)".Unfortunately, without knowing the numbers of cells sorted in each sample, I am failing to see how any of these two reasons argue against it. The olfactory epithelium of young adult mice (8-10 weeks) is populated on average by ~10 million mature OSNs (mOSNs). If each sample contains on average 10,000 sorted cells, that would capture only 0.1% of all mOSNs. If each sample contains 100,000 sorted cells, that would still only make up 1% of all mOSNs, and so on. Since the authors "did not make note of the output cell number" in each sample, how could pooling animals or sorting similar percentages of cells between animals be used as an argument against subsampling? Also, only 4/8 of their samples had between 5-7% of sorted OMP-YFP+ cells, which is not "most samples". I am assuming the authors kept the FACS files, and thus should be able to retrieve those values. Is this the case?

The FACS purification of mOSNs (YFP+ cells) was done by a graduate student, Samantha Jones, who left the laboratory two years ago. Her FACS files only recorded the number of cells used to make the plots we provided in our manuscript (Figure 1—figure supplement 1B). Unfortunately, these FACS files did not indicate the total number of cells sorted and, thus, as we indicated in the previous rebuttal letter, we are sorry but we cannot provide this information. That said, the graduate remembers that she sorted approximately one-half of the dissociated cells that she obtained from dissected olfactory epithelium. While this means that there was some subsampling (as only ~½ of the cells was sorted), nonetheless, a substantial number of mOSNs (YFP+ cells) must have been purified for RNA-seq analysis.

Despite this subsampling, there are several reasons we believe our findings from these FACS-purified cells is valid. First, dissociated OE cells from 3 mice were pooled together to decrease sampling bias (also indicated in our previous rebuttal letter as noted by reviewer 2). This was done to reduce variation caused by differences in individual mice, as well as differences in dissection that might lead to differential recovery of cells in the different “zones” of the olfactory epithelium. In total, we analyzed mOSNs from 24 mice (3 mice x 4 samples x 2 genotyes = 24) for RNA-seq analysis. Second, given that single-cell suspensions were used for FACS sorting (a method that randomly sorts cells), it is unlikely that subsampling would lead to a consistent bias between different samples. Third, confidence in our results comes from the finding that our RNA-seq analysis showed consistent regulation of most of the 78 *Olfr* genes (in several independent *Upf2*-null and control mOSN samples; Figure 1C in the revised manuscript). Fourth, to test the veracity of our RNA-seq analysis, we performed qPCR analysis on 3 of these *Olfr* genes, which verified the regulation of all 3 (Figure 1—figure supplement 2B in the revised manuscript; of note, this figure also shows data from non-*Olfr* genes whose regulation we also verified; this was presented in the previous rebuttal letter). Finally, our single-cell RNA-seq analysis also verified the regulation of these 78 *Olfr* genes (Figure 6H in the revised manuscript). Of note, these OE samples were different than those analyzed by RNA-seq.

In the revised manuscript, we briefly bring up these issues as follows: “A caveat is the OE contains zones enriched for mOSNs expressing particular sets of OLFRs (Miyamichi, Serizawa, Kimura, and Sakano, 2005; Ressler, Sullivan, and Buck, 1994), and thus even though we made an effort to dissect the entire OE for RNA-seq analysis, it is possible that there is zonal heterogeneity in the samples we analyzed. […] Furthermore, our single-cell RNA-seq analysis (which analyzed samples different from those analyzed by RNA-seq) also verified the regulation of these 78 *Olfr* genes (Figure 6H).”.

Finally, what is the logic underlying the following statement: "We note that our manuscript is currently ~30% over the word limit."?

The *eLife* author guidelines states that “…we suggest that authors try not to exceed 5,000 words in the main text…” Our manuscript has >7,000 words.

2.2) The authors here claim that they asked, "what genes are regulated by a particular factor in mOSNs" and that "the depth of RNA-seq analysis we used was sufficient to draw several conclusions". One of the unexpected major conclusions of the authors is that "over half of the downregulated genes in Upf3-null mOSNs are Olfr genes" and state that they "regard this finding as a robust result, as the RNA-seq coverage for Upf3b-null and control OSNs was 95.8% and 96.0%, respectively, of all annotated Olfr gene".Since Olfr and many other genes are expressed zonally in the olfactory epithelium (PMID: 7812149, 15814789, 12709059, 32209480, etc), differences in dissection between animals could easily account for the differences observed in the RNA-seq data from sorted cells. Thus, showing the gene expression distribution of all Olfr genes for each of the sorted samples is a key QC step for this study and it should feature early on in the main figures. The authors should include that in the main text and present a plot containing the mean gene expression values (in log10) for all annotated mouse Olfr in either descending or ascending order for the WT and KO, include similar plots in supplementary for each individual sample and include all the gene names (and not just Ensembl IDs) in the “TPM” sheet of Supplementary file 1 to make this accessible to the community. It would also be very useful to present in supplementary a Venn diagram comparing all the Olfrs expressed in WT vs KO mice.

To address this issue, we examined the expression of all *Olfr* genes in each of our sorted samples (Figure 1—figure supplement 2E in the revised manuscript). This heatmap shows that the individual samples do not show an obvious bias in their expression pattern.

As requested, we also generated a plot showing the Log10-transformed TPM values (mean values) for all *Olfr* genes (Supplementary file 1 in the revised manuscript). A gene list with all annotated *Olfr* genes (in ascending order) was added to the Supplementary file 1. As requested, we also provide a Venn diagram showing the expression pattern of all Olfrs, including the few uniquely expressed in *Upf3b*-null or control mOSNs (Author response image 2). Author response table 3 lists the expression pattern of all these uniquely expressed *Olfrs* in all samples. This table shows that *Olfrs* expressed only in *Upf3b*-null or control mOSNs tend to be lowly expressed; none of these are among the 78 *Olfr* genes we found are statistically downregulated in *Upf3b*-null mOSNs (they were filtered out because of low read count).

**Author response image 2. sa2fig2:** Venn diagram showing the expression of *Olfr* genes in *Upf3b*-null and control mOSNs.

Author response table 3. *Olfr* genes uniquely expressed in *Upf3b*-null and control mOSNs, as determined by our RNA-seq analysis. Reads count are shown.

3) The Materials and methods are now much improved. Only one minor comment here: the authors should include the commercial source of the coyote and bobcat urines.

In the revised manuscript, we provided the commercial source of the coyote urine (Snow Joe) and bobcat urines (Predator Pee).

4) As I mentioned in my original comment, as Gap43 and Krt5 are not expressed in mature OSNs, they should not be called “olfactory neuronal markers”. Instead, Gap43 is a canonical marker for immature neurons and Krt5 a canonical marker for horizontal basal cells.

We have made these corrections.

In line with this, the authors do mention that "mOSN marker gene, Omp, is enriched in these mOSN samples, and that the non-mOSN markers genes, Krt5 and Lgr5, are de-enriched in these samples". Finally, the authors mention that their RNA-seq analysis was done "purified mOSNs, not whole OE". I was indeed aware of this, as it is was already explicit in the manuscript. I suggested plotting something similar to Figure 1C in PMID: 31392275, for 2 main reasons: i) it summarizes in one plot the gene expression levels for the canonical markers for all major cell types populating the olfactory epithelium, and ii) it visually allows the reader assess the levels of “contamination/hitchhiking” by other cell types during the sorting procedure.

In Figure 1—figure supplement 2D we provide a heatmap showing the expression of canonical markers for all major cell types. Most mOSN markers are highly expressed in all 8 samples, and the other markers are more lowly expressed in all 8 samples.

Please correct this by specifically stating the cell types that those genes are indicative of (also, by “de-enriched” I assume the authors mean “depleted”?).

In the revised manuscript, we have corrected these descriptions. We have also changed “de-enriched” to “depleted.”

4.1) The Omp expression data should feature in Figure 1B, as it is the best canonical marker for mOSNs (also see previous response). It is well established that Omp and Gnal are expressed ubiquitously in the majority of mOSNs across all the olfactory epithelium, so I do not see how that could be an explanation.

We moved the *Omp* expression data to Figure 1B. While we agree that it is well established that *Omp* and *Gnal* are ubiquitously expressed in the majority of mOSNs, these two markers may be differentially affected by loss of *Upf3b*, thereby explaining our results. As support, we found that *Upf3b* loss leads to almost complete loss of one mOSN sub-cluster and acquisition of another mOSN sub-cluster, based on our scRNA-seq analysis. The differential expression of *Omp* and *Gnal* could also be because *Upf3b* influences the expression of one of these genes.

4.2) No more comments. This concern is addressed.5) Regarding Figure 6—figure supplement 1B in the revised manuscript: HBCs and GBCs are not OSNs, and it is incorrect to refer them as such. Only immature and mature OSNs should be called OSNs. Please correct this in the figures and throughout the text. Also, on the right-hand panel of Figure 6—figure supplement 1B it would be useful to list all the other cell types that make up 100%.

We thank reviewer 2 for noting this and have made the requisite corrections. We certainly appreciate that HBCs and GBCs are OSN precursors and not yet functional OSNs. With regard to Figure 6—figure supplement 1B, we have made the requested alteration.

6) No more comments. This concern is addressed.7) Olfr genes are expressed zonally in the olfactory epithelium (PMID: 7812149, 15814789, 12709059, 32209480, etc) and differences in dissection between animals could easily translate into differential expression artifacts. Moreover, different mouse strains express different Olfrs at different levels (PMID: 28438259). Since the authors i) have not yet convincingly addressed my previous concerns about a potential bias in dissection (see above), ii) the mice used in this studies were on mixed or different genetic backgrounds, and iii) one of the major claims of the authors is that half of the differentially expressed genes are Olfrs; the authors should validate at least their top 2 differentially expressed Olfrs across the two genotypes.

We apologize for neglecting to indicate in the previous submission that all mice used for the experiments described herein were backcrossed to C57BL/6 for at least 8 passages. We have added this information to the Materials and methods. With regard to the validation requested, qPCR analysis validated the regulation of 3 of 3 *Olfr* genes that we tested, as described above (Figure 1—figure supplement 2B).